# Copper isotopes track the Neoproterozoic oxidation of cratonic mantle roots

Chunfei Chen [1,2] ✉, Stephen F. Foley [2,3], Svyatoslav S. Shcheka [2] & Yongsheng Liu [1] ✉

The oxygen fugacity ($fO_2$) of the lower cratonic lithosphere influences diamond formation, melting mechanisms, and lithospheric evolution, but its redox evolution over time is unclear. We apply Cu isotopes ($\delta^{65}Cu$) of ~1.4 Ga lamproites and < 0.59 Ga silica-undersaturated alkaline rocks from the lithosphere-asthenosphere boundary (LAB) of the North Atlantic Craton to characterize $fO_2$ and volatile speciation in their sources. The lamproites' low $\delta^{65}Cu$ (−0.19 to −0.12‰) show that the LAB was metal-saturated with $CH_4 + H_2O$ as the dominant volatiles during the Mesoproterozoic. The mantle-like $\delta^{65}Cu$ of the < 0.59 Ga alkaline rocks (0.03 to 0.15‰) indicate that the LAB was more oxidized, stabilizing $CO_2 + H_2O$ and destabilizing metals. The Neoproterozoic oxidation resulted in an increase of at least 2.5 log units in $fO_2$ at the LAB. Combined with previously reported high $fO_2$ in peridotites from the Slave, Kaapvaal, and Siberia cratonic roots, this oxidation might occur in cratonic roots globally.

The continental lithosphere–asthenosphere boundary (LAB) is a physically and chemically unique layer[1] and controls and/or modifies the chemical compositions (including volatile abundances) of continental magmas[2]. The oxygen fugacity ($fO_2$) at the cratonic LAB influences diamond formation, melting mechanisms, and lithospheric evolution[2,3]. Generally, the $fO_2$ of the lithospheric mantle decreases with depth at a rate of about 0.4–0.6 log units per GPa because of the increasing stabilization and solubility of $Fe^{3+}$ in garnet with pressure[4,5]. Therefore, the lower reaches of the cratonic lithosphere (>150 km) are reduced with $fO_2 \leq \Delta FMQ-4$ (FMQ: fayalite–magnetite–quartz oxygen buffer)[5]. However, the lower lithosphere has experienced extensive later metasomatism by various types of fluid and/or melt from the asthenosphere throughout the history of the lithosphere mantle[6], which may modify its $fO_2$. Several studies of metasomatized peridotite xenoliths have shown that melt infiltration causes oxidation of the lower cratonic mantle[7–9], but these events are all thought to have occurred shortly before the eruption of their host kimberlites in the Phanerozoic. The oxidation state and redox evolution at the LAB further back in time remain unclear.

Alkaline rocks are generally produced by volatile-triggered melting close to the LAB in continental settings and are more abundant in rifts, where the passage of melts to the surface is easier[2]. The speciation of C–O–H volatiles ($H_2O$, $CO_2$, $CH_4$, and $H_2$) is strongly dependent on the $fO_2$ of the mantle[10,11]. Under reducing conditions, melting is caused by the increased activity of water ($a_{H_2O}$) during the oxidation of $CH_4$[10] and produces melts with high $H_2O$ contents but little carbon (<0.2 wt%) because the solubility of $CH_4$ is much lower than that of $H_2O$ and $CO_2$[12]. Under more oxidized conditions, melting occurs because of further depression of the melting point when $CO_2$ becomes stable[10,13]. The melt compositions and rock types produced depend on the melting conditions and mechanisms (Supplementary Fig. S1). Lamproites are generally silica-saturated mantle magmas formed in reducing conditions in which C–O–H volatiles are dominated by $CH_4$ and $H_2O$[14], as shown by major element and mineral compositions as well as phase stabilities in high-pressure experiments[15,16]. This conclusion is supported by their low carbon contents (generally <0.12 wt%; see the "Methods" section), which is explained by the very low solubility of reduced carbon in melts. In contrast, carbonate-rich and/or silica-undersaturated alkaline rocks, including carbonatites, aillikites,

[1]State Key Laboratory of Geological Processes and Mineral Resources, School of Earth Sciences, China University of Geosciences, 430074 Wuhan, China. [2]School of Natural Sciences, Macquarie University, North Ryde, NSW 2109, Australia. [3]Research School of Earth Sciences, Australian National University, Canberra, ACT 2601, Australia. ✉e-mail: chfchen2016@hotmail.com; yshliu@hotmail.com

kimberlites, and nephelinites are produced in oxidized conditions where C–O–H volatiles are present as $CO_2$ and $H_2O$[13].

The Labrador coast on the North Atlantic Craton (NAC) is a representative occurrence where a sequence of alkaline rocks was emplaced over more than 1200 Ma: lamproites were emplaced in the Mesoproterozoic and carbonate-bearing rocks since the Late Neoproterozoic (Fig. 1a)[17]. The rise of carbonate-bearing magmas since the Late Neoproterozoic in Labrador is consistent with a significant increase in the abundance of carbonatites and kimberlites globally over the past 700 Myr[18,19] (Fig. 1d). This Aillik Bay locality in Labrador experienced three magmatic pulses: lamproites at ca. 1.4 Ga, $CO_2$-rich ultramafic lamprophyres and carbonatites at 590–555 Ma, and silica-undersaturated nephelinites and melilitites at 142 Ma (Fig. 1a). They are believed to be formed by melting of fusible components at the

lithosphere–asthenosphere boundary (see the "Methods" section): phlogopite pyroxenites are required in the source of the lamproites at 5–7 GPa, whereas the later ultramafic lamprophyre (and carbonatite) and nephelinite (and melilitite) periods require phlogopite–carbonate veins at 5–7 GPa, and amphibole wehrlite metasomes at 2–3 GPa, respectively. The lithosphere thinned under Aillik Bay over time from > 180 km for the lamproites and ultramafic lamprophyres/carbonatites to 75–90 km for the nephelinites[17,20,21] (see Supplementary Fig. S2 and see the "Methods" section for geological setting, petrogenesis, and the estimated pressure–temperature of melting).

Cu isotopes ($^{63}Cu$ and $^{65}Cu$) have been widely applied to trace redox reaction processes because significant Cu isotopic variation can occur during oxidation–reduction reactions[22]. They have great potential to characterize the redox state of the mantle sources of these

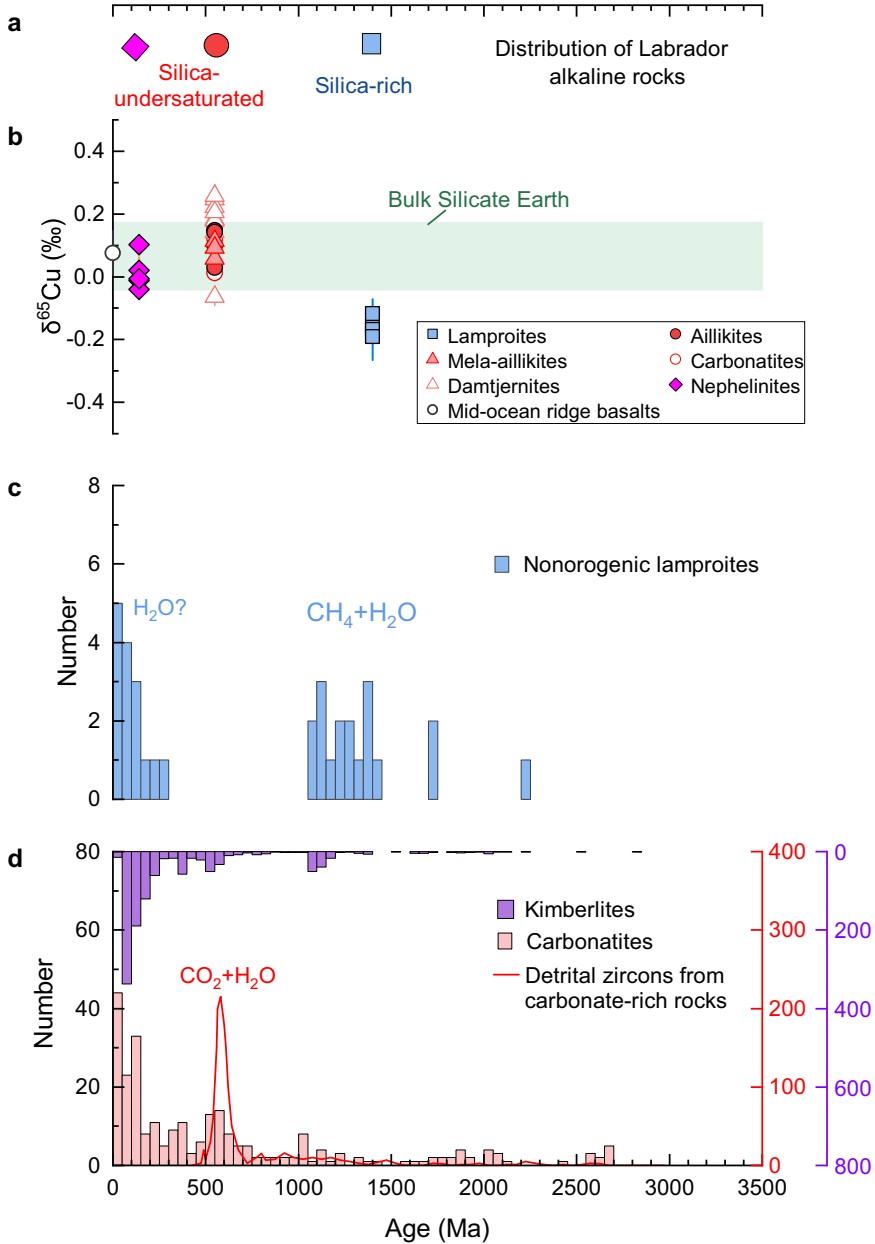

**Fig. 1 | Distribution of alkaline rocks and their Cu isotopes. a** Age distribution of the three stages of rift-related alkaline rocks in Labrador. **b** Cu isotope compositions of rift-related rocks from Labrador in this study. Modern mid-ocean ridge basalts and the bulk silicate Earth are shown for comparison[24]. The error bars for δ⁶⁵Cu values of the alkaline rocks at the three stages and MORBs are 2sd. **c** and

**d** Frequency distribution of ages of global nonorogenic lamproites (Supplementary Data 1), carbonatites[42], and kimberlites[18]. The cumulative U–Pb age data for detrital zircons interpreted to originate from carbonatite-alkaline rocks in Neoproterozoic–Triassic sandstones from Antarctica (red line, $n = 493$)[43] and the C–O–H species in their mantle sources[13,14] are also shown.

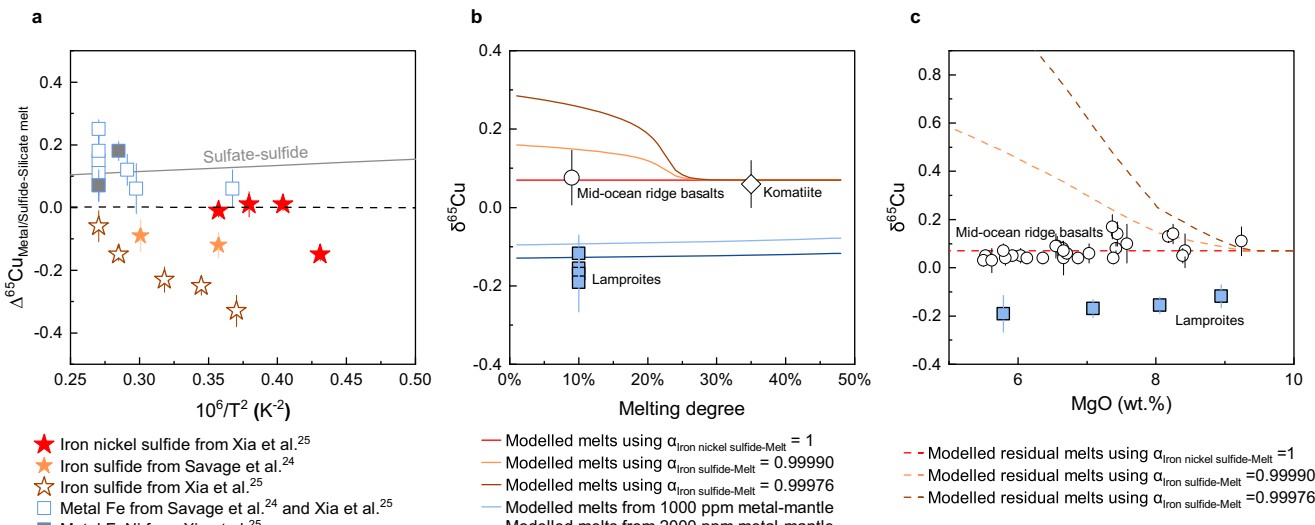

**Fig. 2 | Copper isotope fractionation among phases. a** Cu isotope fractionation between metal (and sulfide) and silicate melt from high-pressure experiments[24,25]. The gray solid line represents Cu isotope fractionation between sulfate and sulfide, extrapolated from calculations in a low-temperature system[22]. **b** Copper isotope compositions of the lamproites from Labrador, average mid-ocean ridge basalts and komatite, and modeled Cu isotope fractionation during partial melting of a peridotite and metal-saturated pyroxenite. Cu isotope fractionation factor $\alpha_{\text{Iron nickel sulfide-Melt}}$ of 1 and $\alpha_{\text{Iron sulfide-Melt}}$ of 0.99976 are from Xia et al.[25], and $\alpha_{\text{Iron sulfide-Melt}}$ of 0.99990 is from Savage et al.[24]. For detailed modeling

calculation (see Supplementary Data 2 and 4). The average melting degrees of the mid-ocean ridge basalts and komatiites are shown for comparison[72,73], and the average melting degree of the Labrador lamproites is assumed to be 10%. **c** Modeled Cu isotope fractionation during magmatic differentiation of a modern mid-ocean ridge basalt-like melt. $\delta^{65}$Cu values of the mid-ocean ridge basalts[24,29,30,74] (MgO > 5 wt%) are shown. The mid-ocean ridge basalts and Labrador lamproites show no relationship between $\delta^{65}$Cu value and MgO content. For detailed modeling calculations, see Supplementary Data 3. The error bars for all $\delta^{65}$Cu values in this figure are 2sd.

alkaline rocks. In a reduced and metal-saturated mantle ($fO_2 \leq$ the iron–wüstite buffer, IW), Cu is mainly sequestered in Fe–Ni alloy as $Cu^0$ and in sulfide as $Cu^{1+}$ [23]. $Cu^0$ in Fe–Ni alloy is more enriched in $^{65}$Cu than $Cu^{1+}$ in silicate melt and sulfide (Fig. 2a)[24,25]. In a metal-free mantle at intermediate oxidation states (IW < $fO_2 \leq$ FMQ), Cu is mainly sequestered in Ni-bearing sulfide, whereas in an oxidized mantle ($fO_2 \geq \Delta$FMQ + 0.5), Cu is mainly hosted by sulfate as $Cu^{2+}$ [26]. Therefore, if a melt is produced with a metal alloy in the source, the copper isotope composition of that melt would be lighter than that of the Bulk Silicate Earth (BSE). Compared to sulfides, the oxidized $Cu^{2+}$ in sulfate is generally enriched in $^{65}$Cu (Fig. 2a)[22].

To delve into the redox evolution at the base of the cratonic lithosphere, we determine the redox state and volatile speciation in the mantle sources of these alkaline rocks from Aillik Bay by employing a novel proxy that copper isotopes (Supplementary Table S1) and compare our results with the redox state recorded in the peridotites from the Slave, Kaapvaal, and Siberia cratons.

## Results and discussions
### Copper isotopic results and effect of magmatic differentiation and crustal recycling
The Cu isotope compositions of the Mesoproterozoic lamproites from Aillik Bay are homogeneous ($\delta^{65}$Cu = −0.12‰ to −0.19‰) and consistently lower than those of modern mid-ocean ridge basalts (MORBs, 0.07 ± 0.1‰)[24] (Fig. 1b). In contrast, the Late Proterozoic ultramafic lamprophyres (aillikites and mela-aillikites) and the Mesozoic nephelinites show MORB-like $\delta^{65}$Cu values (0.03–0.15‰ and 0.04–0.11‰, respectively). Evolved examples of the Late Proterozoic ultramafic lamprophyres (damtjernites) and carbonatites have slightly higher $\delta^{65}$Cu values than the primitive aillikites (Supplementary Fig. S3).

Firstly, it is necessary to consider whether Cu isotopes would fractionate during sulfide segregation from $Cu^{1+}$-bearing silicate melt or during partial melting of a sulfide-bearing mantle in which Cu remains monovalent. High-pressure experiments on Cu isotope fractionation between silicate melt and Ni-free and Ni-bearing sulfides (<1.3

and 21.4–29.0 wt% Ni, respectively)[24,25] show enrichment of $^{63}$Cu in Ni-free sulfides but no resolvable Cu isotopic fractionation between Ni-bearing sulfides and silicate melt (Fig. 2a). Sulfides in natural peridotite and pyroxenite xenoliths have enriched and highly variable Ni contents (5–54 wt%)[27,28] (Supplementary Fig. S4). Although the effect of such a large variation of Ni contents in sulfide on the Cu isotope fractionation factor has not been investigated, the homogeneous $\delta^{65}$Cu values of sulfide-saturated MORBs, sulfide-undersaturated komatiites, and the bulk silicate Earth[24,29] (Supplementary Fig. S5), and the lack of Cu isotope fractionation during the evolution of sulfide-saturated arc magmas[30,31] indicate that most of the sulfide-dominated magmatic processes in the mantle would not cause significant Cu isotope variation in the melts. Highly variable $\delta^{65}$Cu values of some peridotite and pyroxenite xenoliths have been attributed to kinetic Cu isotope fractionation on the scale of individual xenoliths[32], but these effects have not been observed on a large magmatic scale. We model Cu isotope fractionation during magmatic processes using Cu isotope fractionation factors between silicate melt and Ni-free and Ni-bearing sulfides (Fig. 2b, c, see the "Methods" section, Supplementary Figs. S6, S7, and Supplementary Data 2, 3). The results indicate that Ni-bearing sulfide-dominated partial melting and magmatic differentiation would not fractionate Cu isotopes of melts, whereas partial melting and magmatic differentiation involving Ni-free sulfides would result in heavy Cu isotope compositions in the melts (Fig. 2b, c).

Both Mesoproterozoic lamproites and Mesozoic nephelinites show nearly constant Cu contents at 40–80 ppm, independent of MgO content (Fig. 3). This indicates sulfide-saturated evolution, in contrast to the sulfide-undersaturated evolution of komatiites and picrites[33] or oxidized arc magmas[30] with increased Cu contents (Fig. 3). Their Cu isotopes are homogeneous regardless of varying MgO contents, indicating that the effect of magmatic differentiation on $\delta^{65}$Cu is negligible (Fig. 2c). Furthermore, the lamproites and nephelinites have higher Ni contents than MORBs (Supplementary Fig. S4c), suggesting that sulfides participating in partial melting and magmatic differentiation are Ni-bearing rather than Ni-free. This is consistent with the observation

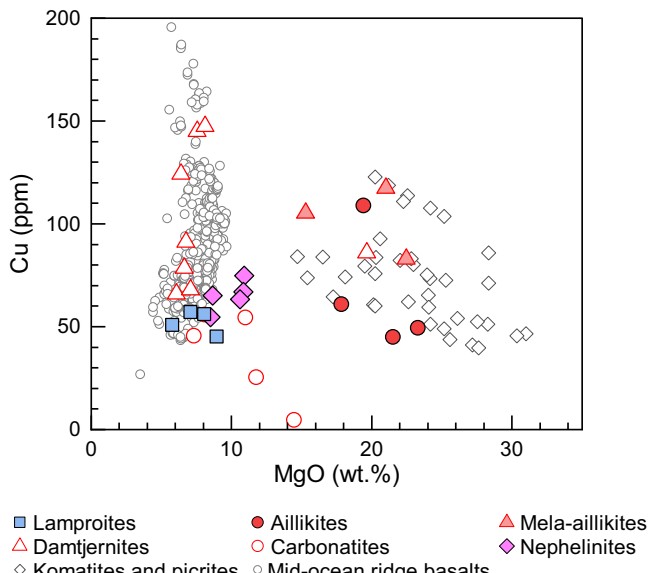

**Fig. 3 | Copper systematics of magmatic rocks.** Cu versus MgO contents in rift-related alkaline rocks from Labrador compared to global mid-ocean ridge basalts[75], komatiites and picrites[52,76,77].

that their Cu isotope compositions plot away from the modeling results of Cu isotope fractionation during Ni-free sulfide-dominated magmatic processes (Fig. 2b, c). Although their enriched trace element and radiogenic isotope compositions have been interpreted to result from metasomatism at the base of the lithosphere mantle caused by melts infiltrating from the asthenosphere[17], sulfide-dominated magmatic processes would not modify their Cu isotope compositions. The primitive Late Proterozoic ultramafic lamprophyres (aillikites and mela-aillikites) show sulfide-undersaturated evolution with increasing Cu contents during melt evolution, consistent with the trend of komatiites and picrites (Fig. 3). Copper behaves as an incompatible element during magmatic differentiation in sulfide-undersaturated magmas, and so Cu isotopes would not be fractionated. The most evolved mela-aillikites show $\delta^{65}Cu$ values similar to the aillikites (Fig. 1b), suggesting that no significant Cu isotope fractionation occurred during magmatic differentiation of the sulfide-undersaturated magmas. This is in agreement with the homogeneous MORB-like $\delta^{65}Cu$ values in sulfide-undersaturated komatiites, picrites[24], and arc magmas[30] (Supplementary Fig. S5). The negligible Cu isotope fractionation during magmatic differentiation of the alkaline magmas discussed above is attributed to the monovalent state of Cu. The evolved damtjernites and carbonatites are thought to have formed by liquid immiscibility from aillikite magma at a shallow crustal depth[20]. The evolved damtjernites and carbonatites show higher and more variable V/Sc ratios than their primary aillikite magmas (Supplementary Fig. S3), which may be a result of variation of magma $fO_2$ during magmatic differentiation. Oxidized $V^{4+}$ (or $V^{5+}$) has a greater affinity for melts than $Sc^{3+}$, and the positive correlation between V/Sc and $\delta^{65}Cu$ values in the evolved damtjernites and carbonatites (Supplementary Fig. S3) reveals oxidation of some $Cu^{1+}$ to $Cu^{2+}$ in the magmas, which is supported by the occurrence of sulfate crystals in carbonatites[20].

Fluids or melts derived from oxidized subducting slabs have higher Cu contents and $\delta^{65}Cu$ values than the mantle[34,35], so the addition of recycled crustal materials could appreciably elevate the Cu content and change the Cu isotopic composition of some mantle rocks[26] (Supplementary Fig. S5). All primitive alkaline rocks have similar or lower Cu content than MORBs (Fig. 3). Furthermore, the addition of high-$\delta^{65}Cu$ crustal materials to the mantle cannot account for the low $\delta^{65}Cu$ values of the lamproites (Supplementary Fig. S5). In

summary, the effect of magmatic differentiation and crustal recycling on Cu isotopes is negligible, so the $\delta^{65}Cu$ of primitive alkaline rocks can be confidently used to decode the redox state of their sources.

## Copper isotopes reveal the oxidation of the LAB under the North Atlantic Craton

The Mesoproterozoic lamproites were produced by the melting of a mantle source involving phlogopite pyroxenite. The low $\delta^{65}Cu$ values of the Mesoproterozoic lamproites could be attributed to either a metasomatic pyroxenite with low $\delta^{65}Cu$ values or isotope fractionation during melting. Melting of a low-$\delta^{65}Cu$ pyroxenite vein would cause melting of and reaction with the surrounding peridotite wall, which could produce a melt with a mixed isotope signature blending characteristics between mantle peridotite with BSE-like $\delta^{65}Cu$ and high $^{143}Nd/^{144}Nd$, and metasomatic pyroxenite with low $\delta^{65}Cu$ and low $^{143}Nd/^{144}Nd$. The lack of correlations between $\delta^{65}Cu$ and Sr-Nd isotopes in the Mesoproterozoic lamproites precludes a low-$\delta^{65}Cu$ pyroxenite source for these lamproites (Supplementary Fig. S4d). As discussed above, Ni-bearing sulfide-dominated magmatic processes could not cause Cu isotope fractionation, also suggested by the BSE-like $\delta^{65}Cu$ values of mantle pyroxenite xenoliths (average $^{65}Cu$ 0.068‰)[36]. Therefore, we suggest that the metasomatic pyroxenites in the source of the lamproites had a BSE-like source $\delta^{65}Cu$ value, and their homogenous and low $\delta^{65}Cu$ values are attributed to the melting of a metal-saturated source as presented below and in Fig. 2b.

Fe-Ni alloy is thought to be stable in the deeper upper mantle[37]. Based on Cu partitioning systematics between Fe−Ni alloy and sulfide at 8 GPa[23], Fe-Ni alloy, along with sulfide, should be the major host of Cu in the metal-saturated mantle. During partial melting of the metal-saturated mantle, sulfide is consumed preferentially relative to Fe−Ni alloy[23]. Consequently, any silicate melts produced are enriched in $^{63}Cu$ and the residue in $^{65}Cu$ due to the high $\delta^{65}Cu$ values of Fe−Ni alloy[24,25]. The low $\delta^{65}Cu$ values of the Mesoproterozoic lamproites (−0.19‰ to −0.12 ‰) appear to require a $Cu^0$-bearing mantle source. We modeled Cu isotope fractionation during partial melting of metal-saturated peridotite with a BSE-like $\delta^{65}Cu$ value of 0.07‰[24] at 6 GPa (details see the "Methods" section, Supplementary Fig. S8, and Supplementary Data 4). The results suggest that melting of peridotite with 1000–2000 ppm Fe−Ni alloy (reasonable Fe−Ni alloy content in metal-saturated peridotite[38]) would produce silicate melt with a $\delta^{65}Cu$ value of −0.1‰ to −0.13‰ (Fig. 2b and Supplementary Fig. S8), which is similar to that of the lamproites. Furthermore, the $\delta^{65}Cu$ value of melt would not vary significantly with melting degree during partial melting of a metal-saturated peridotite. The low and homogenous $\delta^{65}Cu$ values of the lamproites, therefore, indicate that their source (at ≈200 km depth) was metal-saturated at low $fO_2$, which is in keeping with conclusions from geochemical and experimental results[16]. The redox state of the metal-saturated mantle is controlled by the IW buffer reaction (ΔFMQ-5)[39], indicating that the oxygen fugacity of the LAB during Mesoproterozoic was at least as low as IW with $CH_4$−$H_2O$ as the major volatile species at about 6–7 GPa (Fig. 4a).

The MORB-like $\delta^{65}Cu$ values of the Late Proterozoic aillikites and Mesozoic nephelinites suggest that they originated from a sulfide-dominated mantle with neither $Cu^0$−metal nor $Cu^{2+}$−sulfate (IW < $fO_2$ ≤ FMQ). Accordingly, a $CO_2$-rich, oxidized mantle source is needed to produce both the carbonate-rich aillikites at about 5–7 GPa and the nephelinites at 2–3 GPa[10,17]. The mole fraction of $CO_3^{2-}$ in the carbonated melt varies as a function of pressure, temperature, and mantle $fO_2$[40]. At a depth of about 200 km, pure carbonatite melt may exist at ambient mantle $fO_2$ around ΔFMQ-1, with decreasing carbonate components in the melt towards lower $fO_2$[4,40] (Fig. 4a). Using the method of Stagno et al.[40], we estimated the $fO_2$ of the Late Proterozoic primary aillikites equilibrated with the mantle to be between ΔFMQ-2.5 and ΔFMQ-2.2. Given the similar melting depths for the aillikites and lamproites, the LAB under the NAC during the Late Proterozoic was

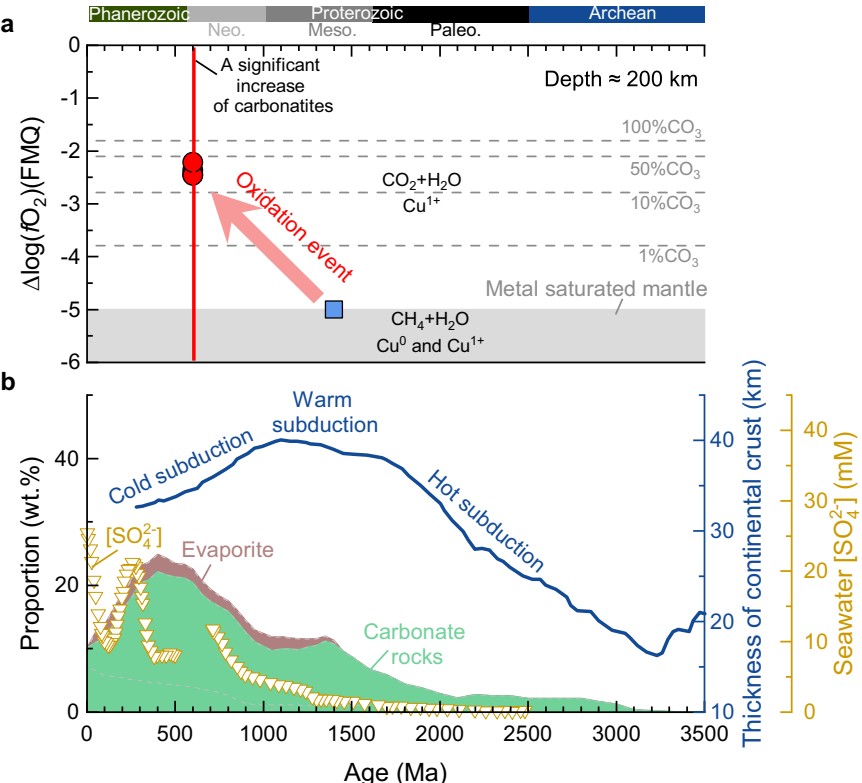

**Fig. 4 | Evolution of $f$O$_2$ of cratonic roots and subduction. a** Redox evolution of the LAB at ≈200 km depth. The gray dotted lines are the oxygen fugacities for melts with different carbonate contents (CO$_3^{2-}$, molar percentage) in equilibrium with diamond-bearing peridotites[40]. The $f$O$_2$ of the LAB of the North Atlantic Craton at a depth of about 200 km did not significantly change since the Late Proterozoic. **b** The proportions of carbonate rocks and evaporites in marine sediments[53] and seawater SO$_4^{2-}$ concentration[78,79] over time. Subduction geotherms are constrained by continental crust thickness[80].

**Fig. 5 | Illustration of mantle oxidation evolution.** The oxidation state of the cratonic mantle over time and the redox melting caused by the change of the speciation of volatiles in the mantle during magmatic episodes in the Mesoproterozoic (**a**) and Late Neoproterozoic (**b**). **a** The $f$O$_2$ of the lithospheric mantle decreases with depth at a rate of about 0.4−0.6 log units per GPa[4,5] until the metal saturation with the nickel precipitation curve[11]. **b** The lithosphere–asthenosphere boundary was significantly oxidized during the Neoproterozoic.

about 2.5 log units more oxidized than during the generation of the lamproites 700−800 Myr earlier in the Mesoproterozoic (Fig. 4a), indicating a significant oxidation event. This is also supported by the fact that the $f$O$_2$ for CO$_2$–H$_2$O fluids in the mantle must be 2−4 log units higher than for CH$_4$–H$_2$O fluid[10].

Therefore, Cu isotopes of the alkaline rocks of the NAC sampled at the three stages of mantle evolution reveal the redox evolution of the LAB. Before the Neoproterozoic, the LAB under the NAC was metal-saturated and reducing, and CH$_4$ and H$_2$O were the dominant C–O–H species (Figs. 4 and 5). Since the Neoproterozoic, the LAB became more oxidized with metal saturation pushed to greater depth (from ≈200 to >250 km)[37] so that CO$_2$ and H$_2$O came to dominate as the dominant C–O–H species (Fig. 5b), and melts infiltrating the lower lithosphere from this time were more oxidized and carbonated. We

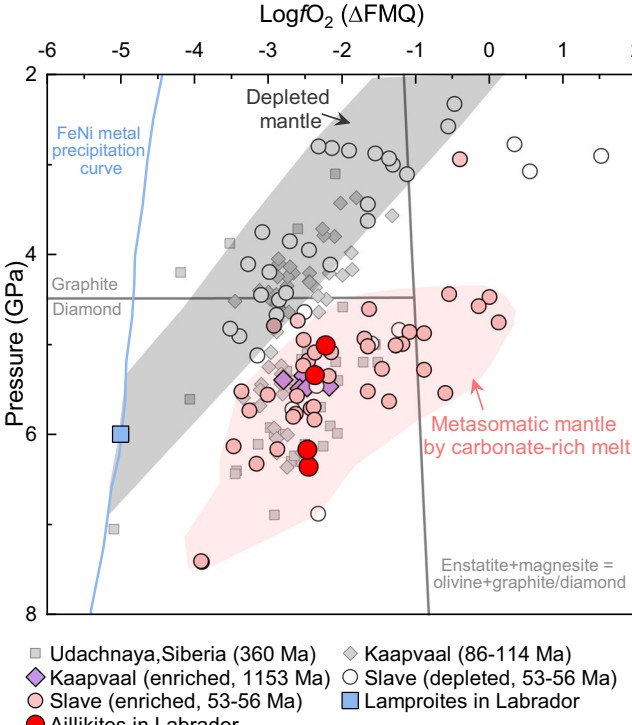

**Fig. 6 | Redox evolution in the lower cratonic lithosphere.** $\Delta \log fO_2$ ($\Delta$FMQ) versus pressure in GPa for the lamproites and aillikites in Labrador, and published peridotite xenolith data from the Slave, Kaapvaal, and Siberian Cratons[8,41,81]. The gray and light red regions are defined by Yaxley et al.[8] and encompass depleted and enriched Slave Craton samples, respectively. The enriched Kaapvaal samples also fall inside the light red field[41]. The gray-filled symbols for the Kaapvaal and Siberia Craton samples are not defined as "depleted" or "enriched" in previous publications. The FeNi metal precipitation curve[11], graphite/diamond transition, and the reaction for carbonate/diamond stability in harzburgitic assemblages (enstatite + magnesite = forsterite + graphite/diamond)[4] are shown for reference. All $fO_2$ for peridotites were calculated by Yaxley et al.[8] and Tappe et al.[41] using the experimental calibration for garnet peridotite assemblages of Stagno et al.[4] and garnet $Fe^{3+}/\Sigma Fe$. The source $fO_2$ of the Labrador lamproites is constrained by metal-saturated conditions at 6 GPa. The source $fO_2$ of the aillikites was calculated using the function of Stagno et al.[40].

suggest that the transition from $CH_4 + H_2O$-dominated and metal-saturated to $CO_2 + H_2O$-dominated and oxidized conditions at the base of the NAC took place in the mantle in Neoproterozoic times prior to 600 Ma (Fig. 4a).

**Neoproterozoic oxidation at the base of cratonic lithosphere**
We compare the redox evolution at the root of the NAC with that of other cratons (Slave, Kaapvaal, and Siberia) as documented by cratonic peridotite xenoliths (Fig. 6). These cratonic peridotite xenoliths have been divided into two groups based on their petrogenesis and equilibrated pressure[8]: (1) reduced and depleted peridotites at low pressures (mostly at <5 GPa) defining the redox state of cratonic lithosphere mantle without modification by mantle metasomatism (gray region in Fig. 6); and (2) relatively oxidized and enriched peridotite xenoliths at pressures >5 GPa (light red region in Fig. 6). Based on the redox evolution with depth recorded in the reduced and un-metasomatized peridotites, the lithosphere mantle would be metal-saturated at the pressure of >5.5–7 GPa if they had not experienced mantle metasomatism by the oxidized agents (Fig. 6). This is consistent with our results on the redox state of the mantle source of the lamproites from the NAC, which in our case is anchored by metal-saturated lithosphere mantle at about 6–7 GPa during the Mesoproterozoic. The extent of

the oxidation event is about 2–3 log units. The increase in $fO_2$ beneath the NAC is similar in magnitude to that documented by the peridotite xenoliths from other cratons. The relatively oxidized and enriched peridotites at >5 GPa from the Slave and Kaapvaal cratons were suggested to result from mantle metasomatism by carbonate-rich melt or fluid[8,41]. Our estimated $fO_2$ for the NAC aillikites is consistent with that of the metasomatic and oxidized peridotite xenoliths from Slave, Kaapvaal, and Siberia (Fig. 6). These results highlight that redox evolution at the root of the NAC is robust and the oxidation of the lithospheric base caused by infiltration of asthenospheric carbonate-rich melts was widespread beneath global cratons.

Our results suggest that the oxidization of the cratonic root of the NAC took place between 1350 and 600 Ma. Peridotite xenoliths from the Premier kimberlite (1150 Ma) in the Kaapvaal craton show that oxidation of the Kaapvaal cratonic root first started prior to 1150 Ma[41]. However, the exact timing of the oxidation of the cratonic roots is poorly constrained because it is difficult to date the carbonatite metasomatism event seen in these peridotite xenoliths. We compiled the global age distribution of nonorogenic lamproites (Supplementary Data 1), kimberlites[18], and carbonatites[42] to decode carbonatite activity in the mantle (Fig. 1c, d). Carbonatites (defined as containing >50% carbonate minerals) and kimberlites show a marked crescendo from the Neoproterozoic onwards[42]. Detrital zircons in sandstones from Antarctica also indicate an onset of widespread carbonatite and silica-undersaturated alkaline magmatism during the Late Neoproterozoic[43] (Fig. 1d), suggesting that the age distribution of carbonatites cannot be simply attributed to preservation bias (see the "Methods" section), as also suggested for kimberlites by Tappe et al.[18]. For the lamproites, we restrict our attention to nonorogenic lamproites because they occur in stable continental regions and are thought to form by melting in reduced conditions[14]. Our compilation shows that the nonorogenic lamproites mainly occurred in the early-middle Proterozoic and Mesozoic–Cenozoic periods, with a conspicuous gap between 1000 and 300 Ma (Fig. 1c).

Based on the distribution of kimberlites and carbonatites, we suggest that the oxidation of the global cratonic roots has continually increased since the Neoproterozoic. The well-known association of carbonate-rich magmatic rocks with rifts since the Late Neoproterozoic[2,44] is thought to be the result of either the remobilization and oxidation of carbon that was stored in the lower lithosphere as diamond[2,6] or by melting of carbonate-bearing metasomatic veins at the LAB with the aid of asthenosphere-lithosphere interaction, indicating that the LAB has been oxidized since the late Neoproterozoic. This is also suggested by a lack of lamproites between 300 and 1000 Ma (Fig. 1a, b). Note that although there has been a resurgence in nonorogenic lamproite activity since 300 Ma (Fig. 1a), isotope data for occurrences on different continents, including Australia[45], North America[46], China[47], and Antarctica[48] indicate that the source components of these lamproites in the lithosphere date from the Mesoproterozoic or earlier. This suggests that these young lamproites are products of the melting of residual and regional Mesoproterozoic veins that pre-date the oxidation of the LAB. We suggest that the oxidation of the global cratonic roots started prior to 1150 Ma, but the large-scale oxidation of the global cratonic roots began in Neoproterozoic times prior to 600 Ma (Fig. 4).

The origin of carbonate-rich melts as metasomatic agents of cratonic roots is the key to understanding the driver of the oxidation of cratonic roots. The continuous increase of carbonatites and kimberlites since 2.0 Ga has been interpreted to be the consequence of secular mantle cooling[18,19], an increasing oxidation state of the asthenosphere through time[49], or increased subduction of oxidized crustal materials into the deep mantle[50]. Secular mantle cooling is not the only factor in controlling the formation of carbonate-rich melts in the deep mantle because oxidized conditions are also required to produce carbonate-rich melts. The $fO_2$ of the asthenosphere was suggested to

have increased by 0.93–1.3 log units at 2.4–1.9 Ga[51,52]. One alternative interpretation is the more effective recycling of oxidized crustal materials ($Fe^{3+}$-rich altered basalt, sulfate, and sedimentary carbonate) into the mantle by colder subduction[53,54] (Fig. 4b). The effective subduction of oxidized crustal materials caused increased redox melting and freezing reactions to produce carbonate-rich melt in the deep asthenosphere[55].

In summary, this study underscores the capability of Cu isotopes to serve as an indicator of melting in the presence of metal alloys, presenting a reliable marker for reduced melts. This could be significant not only for mantle petrology but also for researching reduced bodies within the solar system.

## Methods

### Analytical methods

**Copper isotopes.** Chemical purification and measurement of copper isotopes were performed at the State Key Laboratory of Geological Processes and Mineral Resources, China University of Geosciences, Wuhan, China (SKL GPMR-CUG). The analytical procedures, including sample dissolution, column chemistry, and multi-collector inductively coupled plasma mass spectrometry (MC-ICP–MS) measurements, followed established methods[30,31,56,57]. Aliquots of powdered materials (14-60 mg) were digested by a mixture of concentrated hydrofluoric and nitric acids (HF or $HNO_3$) in Teflon vessels on a hotplate (~120 °C) for 48 h. After digestion, the samples were dried down and re-dissolved in 1 ml HCl and 1 ml $HNO_3$ on a hot plate at 120 °C for 48 h. They were then dried and converted into chloride form by the addition and evaporation of concentrated HCl twice. Finally, the samples were dissolved in 1 ml of 8 mol/l HCl + 0.001% $H_2O_2$ for column chemistry.

The chemical purification method used in this study is modified from Maréchal et al.[58]. Copper purification was performed in a Biorad column containing 2 ml pre-cleaned AG-MP-1M resin. The matrix was removed by elution using 8 ml 8 mol/l HCl + 0.001% $H_2O_2$, after which 28 ml 8 mol/l HCl + 0.001% $H_2O_2$ was used to collect Cu. The column separation procedure was repeated to guarantee the purity of the Cu solution. The Cu fractions were dried down, converted to nitrides, and re-dissolved in 2% $HNO_3$ for isotope analysis. Copper was efficiently separated from the matrix such as Na, K, Ca, Fe, Mg, Si, Ti, Ni, and Mn; for details, see the elution curves for international rock standard BHVO-2 basalt and AGV-2 andesite from Zhu et al.[56] and Liu et al.[57]. High recovery of Cu (>99%) and a low total procedural blank of 3–5 ng were achieved.

Copper isotope ratios were measured with a Nu Plasma 1700 MC-ICP–MS instrument. Isotope measurements were performed using standard-sample bracketing to correct for instrumental mass bias over time. The Cu concentrations of the samples were checked and adjusted to match the Cu standard solution within 10%. The measurements were performed in wet plasma and low-resolution mode. About 300 ppm Cu solution was used for routine analyses and achieved about 4–5 V for $^{63}Cu$. For each measurement, data were collected over three blocks of 25 cycles with 8 s integration. Copper isotope data are reported in standard δ-notation in per mil relative to the standard reference material (SRM) NIST 976.

The geological reference materials BHVO-2 and BIR-1 were digested and analyzed along with samples; the long-term external uncertainty for $\delta^{65}Cu$ (reflected by these reference materials) is better than ±0.05‰ (2sd). The $\delta^{65}Cu$ values obtained for BHVO-2 and BIR-1 are 0.11 ± 0.02‰ (2sd, $n = 3$) and 0.05 ± 0.01‰ (2sd, $n = 3$), respectively, in agreement with previously published data (Supplementary Table S2). All replicates of samples, digested from different aliquots of sample powder, show identical $\delta^{65}Cu$ results with an uncertainty of about 0.05‰ (2sd, Supplementary Table S1).

**Major and trace elements.** No Cu contents have been reported for these alkaline rocks previously, so we have remeasured major and trace elements using solution ICP-MS of the sample rocks (Supplementary Table S1). The sample powders were digested by HF + $HNO_3$ in Teflon bombs and analyzed using an Agilent 7500a ICP-MS at the SKL GPMR-CUG. The detailed sample digestion procedure for ICP-MS analysis is given in Liu et al.[59]. Analyses of rock standards (BCR-2, BHVO-2, and AGV-2) and sample replicates indicate both accuracies and reproducibilities are better than 5% for major elements and 10% for trace elements.

### Geological setting and petrogenesis of the alkaline rocks from the Aillik Bay

**Geological setting and samples.** The NAC was split during the Mesozoic–Cenozoic into two Archean blocks—the Nain Province of Labrador and the Archean terranes of West Greenland, preserving tonalitic crust as old as 3.8 Ga[60] (Supplementary Fig. S1a). The NAC experienced one of the longest craton splitting histories known[17,20]: the first stage corresponds to ca. 1370 Ma olivine lamproite magmas; the second stage corresponds to widespread emplacement of ultramafic lamprophyres and carbonatites in the late Neoproterozoic; the third episode led to successful rifting and the production of new oceanic crust with the eruption of kimberlites, ultramafic lamprophyres and carbonatites, and eventually nephelinite and melilitite magmas along the rift flanks at ca. 200–100 Ma (Supplementary Fig. S1a). The samples selected for Cu isotopic analysis are from the Aillik Bay locality, which is the only known area where alkaline magmatic rocks of all three rifting stages occur. They include four 1.37 Ga old lamproites, 19 ultramafic lamprophyres and carbonatites emplaced between 590 and 555 Ma, and five nephelinites dated at ca. 142 Ma[17,20,21].

**Melting temperature and pressure of the alkaline rocks.** The melting temperature and pressure of the alkaline rocks at Aillik Bay were estimated by Chen et al.[21] using the method of Sun and Dasgupta[61]. This thermobarometer treats the Mg-number (Mg#) of olivine in the mantle source and the $H_2O$ content of primary magmas as key parameters[61]. Chen et al.[21] compiled the Mg# of olivine of carbonated peridotites and wehrlites to calculate the melting temperature and pressure. Chen et al.[21] assumed 80 for Mg# of olivine in the source of the lamproites, and here, we assume 80, 85, and 90 to evaluate the effect of Mg# of olivine on the calculation of melting temperature and pressure (Supplementary Fig. S2). We found that Mg# of olivine has a relatively small effect on the estimation of melting pressure but has a significant effect on the estimation of melting temperature. As shown in Supplementary Fig. S2, the melting pressure was about 5–7 GPa for the lamproites and aillikites, and about 2–3 GPa for the nephelinites. The lithosphere thinned over time from >150 km (5–7 GPa) for the lamproites at ca. 1.4 Ga and ultramafic lamprophyres/carbonatites at 590–555 Ma to 75–90 km (2–3 GPa) for the nephelinites at 142 Ma.

**Metasomatic LAB sources for the alkaline rocks.** The petrogenesis of these alkaline rocks should be considered before decoding the redox state of their sources using Cu isotopes. Cratonic alkaline rocks have been widely thought to be derived from the melting of metasomatized regions of the lower lithospheric mantle (such as the LAB)[17].

However, recent geochemical studies on olivine geochemistry have claimed that cratonic lamproites were formed by the interaction between asthenospheric carbonate-bearing melt and phlogopite-rich wall rocks in the lithosphere mantle, meaning that lamproites would share a similar origin with kimberlites[62,63]. However, this interpretation was based on mineral chemistry alone and overlooks the petrogenesis of many cratonic lamproites, e.g., their low carbon contents (average values calculated as $CO_2$ are 0.05 wt% for Gaussberg, 0.43 wt% for West Kimberley, and 0.5 wt% for Leucite Hills)[64,65]. The "lamproites" studied by Sarkar et al.[62,63] were previously known as orangeites and have much higher $CO_2$ contents (average 5.1 wt%)[65] and distinct origins.

According to the major and trace elements and melting depth estimation of the alkaline rocks at Aillik Bay[17,20,21], we argue that the lamproites originated from melting of the metasomatic lithosphere mantle at the LAB and have no genetic similarity with kimberlites (Supplementary Fig. S1b). The major element compositions of the lamproites closely resemble experimental melts of melting of $CO_2$-free phlogopite pyroxenites. These lamproites are silica-rich, in contrast to the silica-deficient for the aillikites, nephelinites, and kimberlites, as shown in the diagram of CaTs-En join of O'Hara[66] (Supplementary Fig. S1b), which is caused by $CO_2$ in the source.

The estimated melting pressure of the lamproites and aillikites is about 5–7 GPa, similar to those of the cratonic LAB (Supplementary Fig. S2). Therefore, these alkaline rocks at the three stages were formed by the destruction and melting of the metasomatic LAB caused by the convective instabilities in the asthenosphere during passive rifting of the NAC (Supplementary Fig. S2) and can be used to decode the redox state of the metasomatic LAB.

## Effects of low-temperature alteration on Cu isotopes of the studied samples

The lamproite, ultramafic lamprophyre, and nephelinite samples used in this study are fresh with little or no signs of hydrothermal overprinting[20]. For magmatic silicate rocks, homogeneous Cu isotope compositions of both lamproites and nephelinites indicate negligible effects of alteration. The primary magmatic carbonate-bearing aillikites have overlapping mantle-like $\delta^{13}C$ and $\delta^{18}O$ values, and $\delta^{13}C$ and $\delta^{18}O$ values of the evolved damtjernites and carbonatites show a trend of high-temperature fractionation at >600 °C, in contrast to strongly fractionated or hydrothermally overprinted carbonatites[20]. Importantly, the aillikites and mela-aillikites show MORB-like and homogeneous Cu isotope compositions. These observations suggest a negligible effect of alteration on the Cu isotopic compositions of these alkaline rocks.

## Global age distribution of nonorogenic lamproites and carbonatites

We compile the global age distribution of nonorogenic lamproites (Supplementary Data 1 and Fig. 1c, d). The compiled nonorogenic lamproites occur in stable craton regions (Supplementary Data 1) and are much younger than the formation of cratons, indicating that they have no relationship with subduction and collision. These nonorogenic lamproites originated from the lower lithosphere mantle and were formed by melting in reduced conditions[14]. In contrast, orogenic lamproites occur in mobile belts and are generated at shallow depths in post-collisional environments, so they are not relevant for the lower cratonic mantle[67]. The nonorogenic lamproites mainly occurred in the early-middle Proterozoic and Mesozoic–Cenozoic periods, with a conspicuous gap between 1000 and 300 Ma (Fig. 1b).

The age distribution of carbonatites and kimberlites are from Woolley and Kjarsgaard[42] and Tappe et al.[18], respectively. The abundance of the carbonatites and kimberlites show a general increase since ~2 Ga with a significant increase over the past 700 Ma. Previous studies suggested that no relationship exists between the erosion-modified surface areas of kimberlite bodies and their emplacement ages[18], and the distribution of carbonatites exhibits greater variability than the current surface area age distribution[19], suggesting that the global scale 'delayed' appearance of kimberlites and carbonatites is genuine and cannot be simply attributed to preservation bias[18,19]. Furthermore, the detrital zircons in Neoproterozoic–Triassic sandstones from Antarctica also show an onset of widespread carbonatite and silica-undersaturated alkaline magmatism during the Late Neoproterozoic[43] (Fig. 1c), suggesting that the age distribution of carbonatites cannot be simply attributed to preservation bias.

## Modeling of Cu isotope fractionation during magmatic processes

**Partial melting of peridotite at 1.5 GPa.** We quantitatively modeled copper isotope fractionation during the partial melting of peridotite at 1.5 GPa using an incremental batch melting model in which the fractionation factors between melt and residues were calculated at each step (Supplementary Data 2). The starting mineral abundances and melting modes of silicate phases are the same as those in Chen et al.[68]. The starting mineral abundance of sulfide and partition coefficient values (*D*) of Cu between mineral and melt during partial melting are from Lee et al.[69]. The melting of sulfide is dependent of the solubility of sulfur which is a function of temperature, pressure, oxygen fugacity, and FeS activity. The calculation of the solubility of sulfur follows Lee et al.[69]. In detail, the $\delta^{65}Cu$ ratio of starting peridotite is 0.07‰[24]. The oxygen fugacity during partial melting of metal-free peridotite is set at ΔFMQ-1.5. The Cu isotope fractionation factor between sulfide and silicate melt ($\alpha_{Sulfide\text{-}Silicate\ melt}$) is 1 for Ni-bearing sulfide constrained by Xia et al.[25] and is 0.9990 and 0.9975 for Ni-free sulfide from high-pressure experiments of Savage et al.[24] and Xia et al.[25], respectively. The modeling processes and results are presented in Fig. 2b, Supplementary Fig. S6, and Supplementary Data 2.

**Magmatic differentiation of a MORB melt at 0.2 GPa.** We quantitively modeled the behavior of Cu isotopes during magmatic differentiation of a MORB melt at 0.2 GPa using an incremental fractional crystallization calculation in which the fractionation factors between melt and cumulates were recalculated at each step[70]. The crystallization sequence of silicate phases and compositions of melt are calculated using the PETROLOGY software[71] and are the same as those in Chen et al.[70]. The crystallization of sulfide depends on the sulfur content of melt and sulfur concentration at sulfide saturation of the melt. The partition coefficient values of Cu and sulfur concentration at sulfide saturation of the melt are calculated following Lee et al.[69]. The Cu isotope fractionation factors $\alpha_{Sulfide\text{-}Silicate\ melt}$ are the same as those for modeling partial melting of a peridotite as presented above. The modeling processes and results are presented in Fig. 2c, Supplementary Fig. S7, and Supplementary Data 3.

**Partial melting of a metal-saturated pyroxenite at 6 GPa.** We quantitatively modeled Cu isotope fractionation during partial melting of a metal-saturated pyroxenite at 6 GPa using an incremental batch melting model. The abundance of starting minerals and melting of silicate phases are the same as those in Chen et al.[21]. The starting mineral abundance of sulfide, partition coefficient values of Cu, and calculation of the melting abundance of sulfide are the same as those of modeling partial melting of a peridotite as presented above. The melting abundance of sulfide is dependent on the solubility of sulfur in the melt, which is a function of temperature, pressure, oxygen fugacity, and FeS activity. The oxygen fugacity during partial melting is set at ΔFMQ-4. The abundance of Fe–Ni alloy in a metal-saturated peridotite is assumed to be 1000 and 2000 ppm, respectively. The chosen concentrations of 1000-2000 ppm are reasonable for Fe–Ni alloy in metal-saturated peridotite[38]. The partition coefficient of Cu between Fe–Ni alloy and silicate melt is similar to that between sulfide and silicate melt based on experimental data at 8 GPa[23]. As discussed in the main text, the Cu isotope fractionation factor $\alpha_{Sulfide\text{-}Silicate\ melt}$ is 1. The Cu isotope fractionation factor between Fe–Ni alloy and silicate melt ($\alpha_{FeNi\text{-}Silicate\ melt}$) is assumed to be 1.00025 within the range of experimental values of 1.00025–1.00006[24,25]. The modeling processes and results are presented in Fig. 2b, Supplementary Fig. S8, and Supplementary Data 4.

Copper contents in these silicate minerals are too low to affect the behavior of Cu isotopes[69]. In other words, the behavior of Cu isotope fractionation depends on the sulfide and metal regardless of the silicate mineralogy of the source (peridotite or pyroxenite)[69]. With

increasing degree of melting, the consumption of sulfide during partial melting of the metal-saturated pyroxenite at 6 GPa is much slower than that of melting of the peridotite at 1.5 GPa (Supplementary Figs. S6 and S8) due to the much lower solubility of sulfur in the melt at 6 GPa. Furthermore, besides sulfide, the metal is also an important carrier of Cu in the source of the metal-saturated source. Therefore, the Cu isotope fractionation factor between the metal-saturated source and melt would not significantly change with melting degree (Supplementary Data 4), resulting in a small variation of melt $\delta^{65}Cu$ as the increase of melting degree (Fig. 2b and Supplementary Fig. S8).

### Reporting summary

Further information on research design is available in the Nature Portfolio Reporting Summary linked to this article.

### Data availability

The authors declare that all data used in the manuscript are available in Supplementary Tables S1 and S2 and Supplementary Data 1–4 and are also available at https://doi.org/10.6084/m9.figshare.25672236.v1.

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

## Acknowledgements

S.F.F. and C.C. are funded by ARC grant FL180100134. C.C. and Y.L. are funded by NSFC (92355001 and 41530211), the Key R&D Program of China (2019YFA0708400), and SKL-GPMR (MSFGPMR01). Fieldwork in Labrador was supported by Deutsche Forschungsgemeinschaft grant Fo 181/15 to S.F.F. We appreciate the constructive comments from Dr. Sonja Aulbach, Dr. Robert Nicklas, and Dr. Paul Savage that significantly improved the manuscript. We thank Dr. Rong Xu, Yangtao Zhu, Ming Li, and Zongqi Zou for the discussion and their support in the lab.

## Author contributions

C.C. and S.F.F. designed the study. C.C. performed analytical measurements. C.C. wrote the manuscript, and C.C., S.F.F., S.S.S., and Y.L. contributed to interpreting data and revising the manuscript.

## Competing interests

The authors declare no competing interests.
