## [Peer Review File · Nature Communications]

REVIEWER COMMENTS

Reviewer #1 (Remarks to the Author):

This paper reports new geochemical and Cu isotopic data for a suite of alkaline magmas from Aillik Bay (Labrador), which serves as a natural laboratory to investigate the thinning of deep cratonic lithospheric mantle, possibly attended by oxidation, through time. The authors find significantly lower $\delta^{65}\text{Cu}$ values for the older lamproites than for younger, CO_2 -richer magmas and, using modelling in support of their argument, suggest that this reflects that the inferred deep lithospheric sources of these rocks became oxidised by Neoproterozoic time.

I have read this article with great interest and I think it uses a smart and novel approach to gauge $f\text{O}_2$ conditions in the sources of alkaline magmas through time, which has great potential. However, I also see much scope for improvement and would recommend major revisions.

- The authors suggest reduced sources for lamproites and oxidised sources for other alkaline rock types, and the introduction and discussion reveal that this is not something that is unveiled by using Cu isotopes, but which has been suggested before. In this respect I would suggest they change their title to “Cu isotope track the Neoproterozoic oxidation of cratonic mantle roots” or similar, which will take nothing away from the thrust of the paper.

- It has recently been shown that lamproites have a common source in the convecting mantle, but they have stronger interaction with/ contribution from the SCLM (Sarkar et al. 2022 *Geology*, 2023 *JPet*), as true for other alkaline melts like kimberlites (Woodhead et al. 2019 *Nature*; Giuliani et al. 2020 *SciAdv*). In fact, the elephant in the room may be why kimberlites are not included in the analysis? Is it because their composition is dominantly determined in the convecting mantle (e.g., Becker and Le Roex 2006 *JPet* and many others)? A few words would be welcome, as other readers may similarly wonder. Anyways, I feel it should be much more stringently argued, why the analysis presented in this paper is relevant to redox conditions near the LAB, which supposedly (even plausibly) became more oxidised with time. How certain is it that the magma types used for comparison reflect conditions at the LAB, rather than those in the convecting mantle plus some contribution from the SCLM? This needs a much more in-depth analysis, in my opinion, if need be in a supplement.

- Is it sound to take geochemical and Cu isotope results from one region (Aillik Bay) and simply correlate these to all lamproites vs. aillikites/damtjernites/carbonatites/nephelinites globally, as done in Fig. 1 and insinuated by statements in the title and throughout the manuscript, which suggest the authors are proposing a global scenario of deep cratonic SCLM oxidation?

- Fig. 1 could be enhanced – and the authors’ findings supported – by plotting CO_2 contents in the various alkaline magmas of interest as a function of time. This is all the more important as the table reveals that there is quite some overlap between CO_2 contents of Aillik Bay lamproites and younger magma types, which seems inconsistent with a simple reduced vs. oxidised sources scenario.

- This paper aims to investigate oxidation of deep cratonic mantle roots. What is missing is a validation of

the model from actual cratonic mantle samples, which have been entrained prior to the inferred Neoproterozoic oxidation (e.g. Premier/Cullinan in Kaapvaal, Kyle Lake in Superior, and Wajrakarur in Dharwar), and again after this transition (localities world-wide). Even if direct fO_2 estimates are not available, are the xenoliths from older magmas sampling less refertilised (and presumably oxidised) sources than the younger ones, and do the deep xenoliths contain metal-rich “sulphides” that are consistent with the inferred low fO_2 ? Do the xenoliths from younger magmas record fO_2 near the LAB that are permissive of the degree of oxidation shown in Fig. 5B?

- In order to explain the low d_{65Cu} , the authors call for fractionation involving silicate melt and Fe-Ni alloy-bearing mantle that has lost sulphide to melt extraction. There are several possible problems with this scenario: (1) You need to extract some 15-25% of melt to exhaust the sulphide component (depending on initial S content and fO_2 ; Aulbach et al. 2016 RIMG). Given that it is widely agreed that cratonic lithosphere forms by decompression melting (e.g. Herzberg 2004 JPet), this level of depletion seems well plausible for mid-lithospheres but ambitious for SCLM near the LAB, the suggested source of the alkaline melts considered in this paper. The deep SCLM would have experienced the lowest level of depletion. (2) Even so, on sulphide exhaustion, an IPGE-rich alloy will saturate (Mungall and Brenan 2014 GCA) that might also affect Cu isotope partitioning. (3) More importantly, I don't think that alkaline rocks can be formed at melt fractions that are high enough to exhaust sulphide. Is there another scenario that does not require a sulphide-free source, that can be invoked to explain low d_{65Cu} ?

- The explanation for the reappearance of lamproites after the Proterozoic cratonic root oxidation, that they sample older metasomatised sources, is not entirely satisfactory. It requires that these sources, which have low $solidi$, escaped earlier reactivation, and that they also were not affected by the subsequent deep LAB oxidation event. It is not impossible, but amounts to special pleading.

- Statements in the manuscript are often unnecessarily imprecise and sometimes inaccurate, and require more attention to detail. I have indicated these instances directly in the text. A spellcheck and/or attentive reading would have caught many of the typos. Reviewers should not have to fix this many language lapses. There are a number of smaller comments and edits of typos (although I am sure there are more) etc. throughout the word documents of both the main text and the supplementary file, which I have taken the liberty to annotate directly.

In summary, I think a major overhaul is required, but if the authors make the effort to interpret their intriguing Cu isotope data within a solid interpretative framework, it would certainly be of wide interest and appropriate for the journal. I'd be happy to discuss further with the authors and provide clarifications if they so wished. In the meantime, I hope that my comments are useful and that the authors can easily rebut those that are not.

Reviewer #2 (Remarks to the Author):

Review of “Oxidation of cratonic roots during the Neoproterozoic”

This paper utilises a novel isotopic proxy, that of Cu isotopes, to elucidate the oxidation state of the lithosphere-asthenosphere boundary over time, from the Neoproterozoic to modern. By analysing alkali volcanic rocks whose petrogenesis started in the mantle, likely at the LAB, they suggest that increasingly heavy Cu isotope ratios implies that before the Neoproterozoic the lithospheric mantle was reducing (and metallic alloys were stable), and that post-600Ma the lithospheric mantle and the roots of cratons had reached their modern oxidation state. The authors go further to suggest that this is likely linked to the start of “cold subduction”, that is the onset of modern plate tectonics during the Neoproterozoic.

This was a really interesting paper to read – it is well-written and easy to follow (bar the odd spelling mistake) and approaches this topic of the timing of oxidation of cratonic roots from a really novel perspective. I like the idea that Cu isotopes might be useful in this regard, and comment mainly on the authors application of this isotope system as a new proxy for mantle oxidation state. I should state that I'm not an expert in the other evidence for oxidation and timing of oxidation of the cratons, but based on the literature cited in this work, the authors' new results seem to line up well with these previous studies. All in all, I would say that after some adjustments, particularly to the discussion of the application of the Cu isotope proxy and the ruling out of other processes, as well as the modelling, I would be happy to recommend publication of this work.

Before that, though, I would say that one swallow does not make a summer; that is, the anomalously light Cu isotope composition of the Proterozoic lamproite suite measured here is, really, the crux of the argument of this paper. I think a stronger argument might be made if a number of lamproites were analysed, sourced globally, which date from across the Proterozoic - which were all isotopically light wrt MORB/modern carbonatites etc. I don't think it's a reason to reject the paper, but I think the authors might be wary that they might start an arms race for the next people to analyse ancient ultrapotassic samples!

Moving on to the Cu isotope aspect – I have no major reason to question the accuracy of the methods – I note the authors don't seem to cite Marechal et al 1999, which is the original paper that most modern (including this study's) Cu isotope purification methods are based on. One small point is, on line 447, the authors make a point of saying that the method does a good job of separating Ca – why is this particular element picked out? Is it specifically for the carbonatites? Otherwise, normally you would worry about elements like Na, Mg, Ti, Fe, where they have been shown to cause either serious matrix effects during analysis, or generate molecular interferences on the two Cu isotope beams. Because these are weird rocks, with potentially odd matrices that aren't completely comparable to basalts (which are used as the external standards), perhaps explain that matrix contamination in the samples was checked for (I assume it was!).

Finally, the authors mention that their geostandard Cu isotope data match well with previous studies, but do not cite said studies – I agree that their geostandards do match, but only because I analyse Cu isotopes so know the numbers well! Others might not be so familiar. Also, in relation to the precisions quoted for their geostandards, these seem very good compared to the error bars associated with each sample - I guess these errors are from a pooling of a number of measurement data - but then could the authors make this clear?

For the isotopically light lamproite data – I'm not sure, but I don't think the authors spend too much time ruling out mantle metasomatism perhaps altering the Cu isotope composition of the source of these more ancient rocks. I may have missed it, but in that case could the authors make it more clear that the Cu isotope composition of these rocks reflects melting of an "unaltered" mantle composition that has Fe-alloy stable. Could there have been a refertilisation event between 1.4 Ga and 0.6 Ga that affected the mantle in this region?

For the Cu isotope model, which predicts isotopically light melts from a mantle source only when buffered by an Fe-alloy (containing isotopically heavy Cu), I have a couple of questions. The first concerns the assertion that there is no Cu isotope fractionation between sulphides and silicate melt; this is based on the experimental data that show that Ni-rich sulphides and silicates have a negligible Cu isotope fractionation between them. This assertion is used to rule out any Cu isotope fractionation during igneous processes such as sulphide fractionation; although the extant data, which show that MORB and komatiite and also arc lavas all seem to have very predictable Cu isotope compositions seem to suggest that minimal Cu isotope fractionation takes place during sulphide precipitation, I think the authors could have been a bit more careful here when completely ruling this sort of process out.

The first thing to say is that this seems to rely on all sulphides being Ni-rich. This is surely not the case – as the authors know, sulphide petrology and composition can be quite variable, and so simply suggesting that every sulphide involved in the petrogenesis of these rocks is "Ni-rich" is a little difficult to believe based on the evidence provided in this paper; equally, I would ask what "Ni-rich" means – these fractionation factors were taken from a study looking at the crystallisation of the lunar core – are these fractionation factors applicable to Earth's mantle? This criticism may seem harsh, but all I am saying is that Ni-poor sulphides can fractionate Cu isotopes when equilibrating with a silicate melt – hence I would recommend the authors at least mention this, or take this into account, in their models. I'm pretty sure, when I've modelled mantle melting and sulphide precipitation, you don't get a huge difference in Cu isotope composition in melts, as the maximum sulphide-silicate fractionation factors are pretty small, especially at mantle temperatures.

In relation to the authors' metal-sulphide fractionation models, these are a little difficult to follow, and they aren't described in much detail – the authors mainly refer to their supplementary information database. I think a little more detail on the models in the written component of the supplementary material, plus perhaps show how [Cu] varies in the melts and restite in the models – as this can help the reader in following the model graphically (as it is, the figure displaying the model is not intuitive because of the buffering of the sulphide in the melt by the metal phase).

Finally, could the authors comment on how not all sulphides in mantle rocks are available to melt as well as being less susceptible to metasomatism, i.e. if these are present as inclusions in olivine – is there a potential for a "hidden reservoir" of Cu isotopes in mantle rocks here.

Minor comments

On line 80, the authors state that the Methods contain sample descriptions and petrology – but really this is limited to the thermobarometry. I think it would be good to include at least a brief description of the rocks utilised in this work, in terms of where and when they were collected, what their mineralogies

are, etc.

There seems to be a lot of assumptions/generalisations used in the thermobarometric calculations – whilst I'm sure these are all strong assumptions, it might be worth the authors including some statements to explain how sensitive their P and T estimates are to these assumptions.

In figure 4, that authors have a label that states that global plate tectonics starts at 3.2 Ga. Whilst this is one of the periods that is commonly accepted that plate tectonics started globally, perhaps they authors might just take this off, as this is still a date that is up for debate.

I'm not sure that figure 5 in the main text really is adding much to the piece – I think showing an updated Figure S5, which includes [Cu] in the melt and restite, would be more instructive.

Reviewer #3 (Remarks to the Author):

The study of Chen et al. (2023) seeks to constrain the redox evolution of the LAB from the Proterozoic to the present-day using Cu isotopic data from a suite of alkaline rocks located in the NAC of Labrador, Canada. They find a temporal shift in Cu isotopic data that demonstrates that the LAB at this locality has oxidized in the mid-Proterozoic, possibly as a result of the rise of cold, modern style subduction. The premise and topic of this study are both interesting and clever, and for that reason it deserves to be published in this journal, as mantle redox evolution in all its permutations has strong effects on atmospheric evolution. The use of Cu isotopes to quantify mantle redox at the LAB is both innovative and interesting.

However, currently the paper is bogged down by a variety of ideas and exercises that prevent a clear story about Cu isotopes from being imparted to the reader. Chiefly among these is the use of a compilation of a very low number (31 by my counting) of nonorogenic lamproites and a much larger number of carbonatites to postulate a switch from predominant lamproites to predominant carbonatite production. Given the very general title of the paper, the reader is confused and initially thinks that the paper doesn't focus on Cu isotopes of a single locality, and is instead concerned with compiling ages of alkali rocks. The discussion of the age distributions of alkali rocks is treated like a headliner of the paper but it is not, and it should be relegated to the discussion section where the authors try to link the Ailak locality with global change. This brings me to my second overarching problem with the paper: the generalization of a single locality to the whole Earth and the lack of discussion of the NAC specifically in favor of assuming that it reflect global trends. The mantle lithosphere shows immense amounts of redox heterogeneity relating to metasomatism, and a global secular trend is not demonstrated by a trend in a single 20x20 km area. If the authors were to alter the title and discussion to instead discuss how their data may be indicative of a global change as opposed to definitively stating that these data show global change, the manuscript would be much stronger. The method used here (Cu isotopes of alkali rocks) can be concluded to show promise for quantifying global trends once more localities are analyzed, and that in my opinion is the most important conclusions of the paper.

Two more general points I have with the study: how easy is it to distinguish orogenic from nonorogenic

lamproites in tectonically complex Proterozoic terranes? Is it possible that misidentification could have led to the distribution in nonorogenic lamproites observed? Finally, much is made of the globally rare nature of carbonatites in the Precambrian, but the low preservation potential of carbonatites in the rock record is never once mentioned. This is an oversight and should be corrected, as given the low amount of crust available from the early Earth and the rarity of carbonatites in general throughout Earth history, the exponential increase curve of carbonatites shown must be placed in proper context.

Overall, the paper is well written and the figures are thorough and highly professional. Re-focusing the paper on Cu isotopic data for NAC lavas and backing away from global generalizations and the age compilation data will serve this excellent dataset better.

Specific Comments

-In this section I detail specific in-line comments. I have also provided numerous grammatical and usage edits in the "tracked changes" file returned with this review.

-In Line 135, V/Sc is used as a proxy for oxygen fugacity of low melt fraction, enriched lavas. V/Sc is only a useful oxybarometer if three conditions are met: 1) melting does not leave garnet in the residue or fractionate garnet, due to the strongly compatible nature of Sc in garnet (much like the HREE) relative to V, 2) the degree of partial melting is well constrained and similar between the two types of rocks being compared. At low degrees of partial melting (<10%) like those examined here, small changes in the degree of partial melting can strongly influence V/Sc at constant fO_2 (see Figure 1 of Li and Lee, 2004), 3) Finally and most importantly, V/Sc is a forward model for fO_2 and assumes a constant V/Sc in the mantle source regions of the rocks examined. This is almost certainly untrue in this case as the mantle source regions are highly metasomatized lithosphere. I would not use V/Sc as a redox proxy at all, as your Cu isotope redox story is compelling enough without it.

-In Line 145 you use Cu/Sc to argue that mantle source regions do not show prior Cu enrichments. You use the observation that Cu/Sc is elevated in OIB relative to MORB to invoke Cu enrichment in the OIB source mantle, but similar to my previous comment, this is almost certainly the result of the compatibility of Sc in garnet relative to Cu and unrelated to Cu enrichment, as garnet is found in the residue of most OIB but not MORB. If you want a trivalent cation that is less influenced by garnet, try using Ga, as V/Ga has been suggested for use instead of V/Sc for this very reason (Laubier XXX). Alternatively, all of the V/Sc and Cu/Sc discussion can be cut and the manuscript focused better on the main point: Cu isotopes.

-Line 159: How did you choose your starting isotopic value? Is it the accepted BSE value? If so cite it.

-Line 178: What is NAC? Define terms appropriately.

-Line 182: What about the youngest rocks and their fO_2 ? Tell the three stages of the story here, not just the first two.

Line 191: What about preservation bias of carbonatite rocks? This is never mentioned or addressed. Carbonatite rocks can weather much more readily than silicates and therefore their preservation from the Archean to present day is perhaps unlikely.

-Line 222: This almost certainly testably by examining initial Nd and Sr isotopic signatures of global carbonatites and lamproites, as subducted recycled materials should impart a strongly enriched signature. I am not suggesting that you go through the exercise of compiling these data and seeing if such a shift is evident, but I am saying you should state that this would be a good way to test your model in a future study. Besides it is always good practice to end a paper in an open-ended manner, with an eye towards what sort of future data or studies could support the model presented.

-Figure 4: Take out “vast” from the carbonatite-in line

-Figure 4: Why is the “onset of plate tectonics” marked on this figure? This is never discussed in the text and remains a hotly debated topic. This figure is busy, and this label adds little to it. As you only focus on the Proterozoic to Phanerozoic, why not start the figure at 2500 Ma?

-Figure 5: this is an excellent figure for illustrating your model for LAB, good job.

-Line 532: Which is it? 1000 or 2000 ppm or did you vary the concentration between these two values?

Response to referees' comments, manuscript NCOMMS-23-32963

Chunfei Chen, Stephen F. Foley, Svyatoslav S. Shcheka, Yongsheng Liu

"Copper isotopes track the Neoproterozoic oxidation of cratonic mantle roots"

Referees' comments in black

Replies in blue

We thank the referees for the constructive comments and suggestions that significantly improved the manuscript.

REVIEWER COMMENTS

Reviewer #1 (Remarks to the Author):

This paper reports new geochemical and Cu isotopic data for a suite of alkaline magmas from Aillik Bay (Labrador), which serves as a natural laboratory to investigate the thinning of deep cratonic lithospheric mantle, possibly attended by oxidation, through time. The authors find significantly lower $\delta^{65}\text{Cu}$ values for the older lamproites than for younger, CO_2 -richer magmas and, using modelling in support of their argument, suggest that this reflects that the inferred deep lithospheric sources of these rocks became oxidised by Neoproterozoic time.

I have read this article with great interest and I think it uses a smart and novel approach to gauge $f\text{O}_2$ conditions in the sources of alkaline magmas through time, which has great potential. However, I also see much scope for improvement and would recommend major revisions.

- The authors suggest reduced sources for lamproites and oxidised sources for other alkaline rock types, and the introduction and discussion reveal that this is not something that is unveiled by using Cu isotopes, but which has been suggested before. In this respect I would suggest they change their title to "Cu isotope track the Neoproterozoic oxidation of cratonic mantle roots" or similar, which will take nothing away from the thrust of the paper.

Reply: We have modified the title to "Copper isotopes track the Neoproterozoic oxidation of cratonic mantle roots".

- It has recently been shown that lamproites have a common source in the convecting mantle, but they have stronger interaction with/ contribution from the SCLM (Sarkar et al. 2022 *Geology*, 2023 *JPet*), as true for other alkaline melts like kimberlites (Woodhead et al. 2019 *Nature*; Giuliani et al. 2020 *SciAdv*). In fact, the elephant in the room may be why kimberlites are not included in the analysis? Is it because their composition is dominantly determined in the convecting mantle (e.g., Becker and Le Roex 2006 *JPet* and many others)? A few words would be welcome, as other readers may similarly wonder. Anyways, I feel it should be much more stringently argued, why the analysis presented in this paper is relevant to redox conditions near the LAB, which supposedly (even plausibly) became more oxidised with time. How certain is it that the magma types used for comparison reflect conditions at the LAB, rather than those in the convecting mantle plus some contribution from the SCLM? This needs a much more in-depth analysis, in my opinion, if need be in a supplement.

Reply: We have added a section (Geological setting and petrogenesis of the alkaline rocks from the Aillik Bay) in the "Methods" (Lines 621-646) and a figure of the CaTs-En join of O'Hara¹ (Supplementary Fig.

S1b) to discuss the petrogenesis of the alkaline rocks from Aillik Bay.

The recent geochemical studies on olivine geochemistry by Sarkar and others interpreted cratonic lamproites to be formed by the interaction between asthenospheric carbonate-bearing melt and phlogopite-rich wall rocks in the lithosphere mantle, and therefore that they are related to kimberlites^{2,3}. The rocks they consider used to be known as orangeites, but have been renamed by various geochemists in the last few years as “carbonate-rich lamproites” or “lamproites var. Kaapvaal”. These conclusions appeal to mineral chemistry but ignore the petrogenesis. Most cratonic lamproites are extremely poor in carbon (see Figure below, noting log scale for CO₂), which is consistent with a reduced source, because the solubility of reduced carbon in melts is very low. An asthenospheric carbonate-rich melt could never achieve the exceptionally low CO₂ contents of most lamproites. Our interpretation is that the re-assignment of orangeites as lamproites is a mistake and results only in confusion.

According to the major and trace elements and melting depth estimation of the Aillik Bay alkaline rocks⁴⁻⁶, we suggest that the lamproites originated from the melting of the metasomatic lithosphere mantle at LAB and do not have any genetic relationship with kimberlites. Firstly, the major element compositions of the lamproites highly resemble the experimental melts of melting of CO₂-free phlogopite pyroxenites. These lamproites are silica-saturated (Supplementary Fig. S1b), in contrast to the silica-deficiency (caused by CO₂) for the aillikites, nephelinites, and kimberlites. Secondly, the estimated melting pressure of the lamproites and aillikites is about 5-7 GPa, similar to those of the cratonic LAB (Supplementary Fig. S2). Therefore, these alkaline rocks at the three stages were formed by the destruction and melting of the metasomatic LAB during passive rifting of the NAC (Supplementary Fig. S2), and can be used to decode the redox state of the metasomatic LAB.

Figure caption: SiO₂ versus CO₂ contents in lamproites and orangeites.

- Is it sound to take geochemical and Cu isotope results from one region (Aillik Bay) and simply correlate these to all lamproites vs. aillikites/damjtjernites/carbonatites/nephelinites globally, as done in Fig. 1 and insinuated by statements in the title and throughout the manuscript, which suggest the authors are proposing a global scenario of deep cratonic SCLM oxidation?

Reply: We have reworked the section on global extrapolation to add more explanation (Lines 209-228). We

compare the redox evolution at the root of the NAC with those of other cratons (Slave, Kaapvaal, and Siberia) as documented by cratonic peridotite xenoliths (Fig. 6). These cratonic peridotite xenoliths have been divided into two groups based on their petrogenesis and equilibrated pressure⁷: (1) reduced and depleted peridotites at low pressures (mostly at <5 GPa) defining the redox state of a craton lithosphere mantle without modification from mantle metasomatism (grey region in Fig. 6); and (2) relatively oxidized and enriched peridotite xenoliths at the pressure of >5 GPa (light red region in Fig. 6). Based on the redox evolution with depth recorded in the reduced and un-metasomatic peridotites, the lithosphere mantle would be metal-saturated at the pressure of >5.5-7 GPa if they did not experience mantle metasomatism by the oxidized agents (Fig. 6). This is consistent with our result on the redox state of the mantle source of the lamproites from the NAC, indicating a metal-saturated lithosphere mantle at about 6-7 GPa during the Mesoproterozoic. The relatively oxidized and enriched peridotites at > 5 GPa from the Slave and Kaapvaal cratons were suggested to result from mantle metasomatism by carbonate-rich melt/fluid^{7,8}. Our estimated fO_2 of the aillikites from the NAC in this study is consistent with these of the metasomatic and oxidized peridotite xenoliths from Slave, Kaapvaal, and Siberia (Fig. 6). These highlight that our results of redox evolution at the root of the NAC are robust and the oxidization of the lithospheric base caused by infiltration of asthenospheric carbonate-rich melts occurred beneath the global cratons.

- Fig. 1 could be enhanced – and the authors’ findings supported – by plotting CO₂ contents in the various alkaline magmas of interest as a function of time. This is all the more important as the table reveals that there is quite some overlap between CO₂ contents of Aillik Bay lamproites and younger magma types, which seems inconsistent with a simple reduced vs. oxidised sources scenario.

Reply: The CO₂ values in the lamproites do not represent CO₂ in the magma because these values only represent carbon contents and we cannot measure the redox state of carbon in the samples. We have changed the reported data from CO₂ to carbon content.

We have plotted our samples in the diagram of the CaTs-En join of O’Hara¹, showing that the aillikites and nephelinites are extremely silica-deficient (caused by CO₂) whereas the lamproites are in the same region as experimental melts from CO₂-free pyroxenites.

- This paper aims to investigate oxidation of deep cratonic mantle roots. What is missing is a validation of the model from actual cratonic mantle samples, which have been entrained prior to the inferred Neoproterozoic oxidation (e.g. Premier/Cullinan in Kaapvaal, Kyle Lake in Superior, and Wajrakarur in Dharwar), and again after this transition (localities world-wide). Even if direct fO_2 estimates are not available, are the xenoliths from older magmas sampling less refertilised (and presumably oxidised) sources than the younger ones, and do the deep xenoliths contain metal-rich “sulphides” that are consistent with the inferred low fO_2 ? Do the xenoliths from younger magmas record fO_2 near the LAB that are permissive of the degree of oxidation shown in Fig. 5B?

Reply: The answers to this are the same as for the above comment. We have compared our results of redox evolution at the root of the NAC with those of other cratons (Slave, Kaapvaal, and Siberia) as revealed by the cratonic peridotite xenoliths. Our results are consistent with those from peridotite xenoliths from Slave, Kaapvaal, and Siberia.

The peridotite xenoliths from old kimberlite magmas also experienced metasomatism by carbonate-bearing melts or kimberlites⁸. Based on the evidence recorded in the peridotites, the lithosphere would be metal-saturated at > 5.5-7 GPa if they did not experience mantle metasomatism by oxidized agents (Fig. 6). This is consistent with our result on the redox state of the mantle source of the lamproites from the NAC,

indicating a metal-saturated lithosphere mantle at about 6-7 GPa during the Mesoproterozoic.

- In order to explain the low $\delta^{65}\text{Cu}$, the authors call for fractionation involving silicate melt and Fe-Ni alloy-bearing mantle that has lost sulphide to melt extraction. There are several possible problems with this scenario: (1) You need to extract some 15-25% of melt to exhaust the sulphide component (depending on initial S content and $f\text{O}_2$; Aulbach et al. 2016 RIMG). Given that it is widely agreed that cratonic lithosphere forms by decompression melting (e.g. Herzberg 2004 JPet), this level of depletion seems well plausible for mid-lithospheres but ambitious for SCLM near the LAB, the suggested source of the alkaline melts considered in this paper. The deep SCLM would have experienced the lowest level of depletion. (2) Even so, on sulphide exhaustion, an IPGE-rich alloy will saturate (Mungall and Brenan 2014 GCA) that might also affect Cu isotope partitioning. (3) More importantly, I don't think that alkaline rocks can be formed at melt fractions that are high enough to exhaust sulphide. Is there another scenario that does not require a sulphide-free source, that can be invoked to explain low $\delta^{65}\text{Cu}$?

Reply: We present the details of modelling Cu isotope fractionation during partial melting of a metal-saturated mantle at 6 GPa and compare the results with modelling Cu isotope fractionation during partial melting of a peridotite at 1.5 GPa. The details can be found in the section "Modelling of Cu isotope fractionation during various magmatic processes" of "Methods" (Lines 678-731), Fig. 2b-c, Supplementary Figs. S6 and S8, and Supplementary Data S2 and S4.

Our results indicate low $\delta^{65}\text{Cu}$ values for the melts once the melting of a metal-saturated mantle is triggered. Copper isotope compositions of the melt depend on the Cu isotope fractionation factor between the metal-saturated source and melt ($\alpha_{\text{Source-Melt}}$). Due to heavy isotopes in the metal of the mantle source, the Cu isotope fractionation factor $\alpha_{\text{Source-Melt}} > 1$, and the produced melts have low $\delta^{65}\text{Cu}$ value.

With increasing melting degree, the consumption of sulfide during partial melting of the metal-saturated pyroxenite at 6 GPa is much slower than that of melting of the peridotite at 1.5 GPa (Supplementary Figs. S6 and S8) due to the much lower solubility of sulfur of melt at 6 GPa. Furthermore, besides sulfide, the metal is also an important carrier of Cu in the source of the metal-saturated source. Therefore, the Cu isotope fractionation factor between the metal-saturated source and melt would not significantly change with melting degree (Supplementary Data S4), resulting in a small variation of melt $\delta^{65}\text{Cu}$ as the increase of melting degree (Fig. 2B and Supplementary Fig. S8).

The low $\delta^{65}\text{Cu}$ values of the lamproites do not require sulfide exhaustion during melting. Although we do not know the melting degree for the lamproites, it is possible to melting of the 15 wt.% of a mixed mantle source with fusible metasomatic component (not peridotite) at the LAB⁹.

- The explanation for the reappearance of lamproites after the Proterozoic cratonic root oxidation, that they sample older metasomatised sources, is not entirely satisfactory. It requires that these sources, which have low solidi, escaped earlier reactivation, and that they also were not affected by the subsequent deep LAB oxidation event. It is not impossible, but amounts to special pleading.

Reply: This is only the case if oxidation events are uniform over the whole area of the craton base, whereas they are localised influxes of melts. Nevertheless, we have weakened the reliance on evidence from the distribution of lamproites.

We have emphasized the distribution of carbonatite and kimberlite activities to roughly constrain the time of oxidization of cratonic root.

- Statements in the manuscript are often unnecessarily imprecise and sometimes inaccurate, and require

more attention to detail. I have indicated these instances directly in the text. A spellcheck and/or attentive reading would have caught many of the typos. Reviewers should not have to fix this many language lapses. There are a number of smaller comments and edits of typos (although I am sure there are more) etc. throughout the word documents of both the main text and the supplementary file, which I have taken the liberty to annotate directly.

Reply: We have thoroughly revised the manuscript.

In summary, I think a major overhaul is required, but if the authors make the effort to interpret their intriguing Cu isotope data within a solid interpretative framework, it would certainly be of wide interest and appropriate for the journal. I'd be happy to discuss further with the authors and provide clarifications if they so wished. In the meantime, I hope that my comments are useful and that the authors can easily rebut those that are not.

Reply: Thanks for your constructive suggestions and we have made a major overhaul on the manuscript. Hopefully, these modifications have clarified these issues.

Reviewer #2 (Remarks to the Author):

Review of "Oxidation of cratonic roots during the Neoproterozoic"

This paper utilises a novel isotopic proxy, that of Cu isotopes, to elucidate the oxidation state of the lithosphere-asthenosphere boundary over time, from the Neoproterozoic to modern. By analysing alkali volcanic rocks whose petrogenesis started in the mantle, likely at the LAB, they suggest that increasingly heavy Cu isotope ratios implies that before the Neoproterozoic the lithospheric mantle was reducing (and metallic alloys were stable), and that post-600Ma the lithospheric mantle and the roots of cratons had reached their modern oxidation state. The authors go further to suggest that this is likely linked to the start of "cold subduction", that is the onset of modern plate tectonics during the Neoproterozoic.

This was a really interesting paper to read – it is well-written and easy to follow (bar the odd spelling mistake) and approaches this topic of the timing of oxidation of cratonic roots from a really novel perspective. I like the idea that Cu isotopes might be useful in this regard, and comment mainly on the authors application of this isotope system as a new proxy for mantle oxidation state. I should state that I'm not an expert in the other evidence for oxidation and timing of oxidation of the cratons, but based on the literature cited in this work, the authors' new results seem to line up well with these previous studies. All in all, I would say that after some adjustments, particularly to the discussion of the application of the Cu isotope proxy and the ruling out of other processes, as well as the modelling, I would be happy to recommend publication of this work.

Reply: Thanks for your constructive comments and suggestions.

Before that, though, I would say that one swallow does not make a summer; that is, the anomalously light Cu isotope composition of the Proterozoic lamproite suite measured here is, really, the crux of the argument of this paper. I think a stronger argument might be made if a number of lamproites were analysed, sourced globally, which date from across the Proterozoic - which were all isotopically light wrt MORB/modern

carbonatites etc. I don't think it's a reason to reject the paper, but I think the authors might be wary that they might start an arms race for the next people to analyse ancient ultrapotassic samples!

Reply: We have weakened the reliance on evidence from the distribution of lamproites. We have compared our results with the redox state of cratonic peridotite xenoliths (Lines 209-228). See responses to Reviewer 1 on this topic.

Moving on to the Cu isotope aspect – I have no major reason to question the accuracy of the methods – I note the authors don't seem to cite Marechal et al 1999, which is the original paper that most modern (including this study's) Cu isotope purification methods are based on. One small point is, on line 447, the authors make a point of saying that the method does a good job of separating Ca – why is this particular element picked out? Is it specifically for the carbonatites? Otherwise, normally you would worry about elements like Na, Mg, Ti, Fe, where they have been shown to cause either serious matrix effects during analysis, or generate molecular interferences on the two Cu isotope beams. Because these are weird rocks, with potentially odd matrices that aren't completely comparable to basalts (which are used as the external standards), perhaps explain that matrix contamination in the samples was checked for (I assume it was!).

Reply: We have added more details and references to describe the chemical purification method. The chemical purification method in this study is modified from Maréchal et al.¹⁰. Copper was efficiently separated from the matrix such as Na, K, Ca, Fe, Mg, Si, Ti, Ni, Mn, and the details see the elution curves for international rock standard BHVO-2 basalt and AGV-2 andesite from Zhu et al.¹¹ and Liu et al.¹². High recovery of Cu (>99%) and a low total procedural blank of 3-5 ng were achieved.

Finally, the authors mention that their geostandard Cu isotope data match well with previous studies, but do not cite said studies – I agree that their geostandards do match, but only because I analyse Cu isotopes so know the numbers well! Others might not be so familiar. Also, in relation to the precisions quoted for their geostandards, these seem very good compared to the error bars associated with each sample - I guess these errors are from a pooling of a number of measurement data - but then could the authors make this clear?

Reply: We have added a supplementary table (Supplementary Table S2) to compare our results of reference standards with those in previous studies.

For the isotopically light lamproite data – I'm not sure, but I don't think the authors spend too much time ruling out mantle metasomatism perhaps altering the Cu isotope composition of the source of these more ancient rocks. I may have missed it, but in that case could the authors make it more clear that the Cu isotope composition of these rocks reflects melting of an “unaltered” mantle composition that has Fe-alloy stable. Could there have been a refertilisation event between 1.4 Ga and 0.6 Ga that affected the mantle in this region?

Reply: The Mesoproterozoic lamproites were produced by melting of a mantle source involving phlogopite pyroxenite. The low $\delta^{65}\text{Cu}$ values of the Mesoproterozoic lamproites could be conceived as being due to either a metasomatic pyroxenite with low $\delta^{65}\text{Cu}$ values or isotope fractionation during melting. Melting of a mantle source involving $\delta^{65}\text{Cu}$ -low pyroxenite could produce a melt with mixed isotopes between mantle peridotite and pyroxenite. No correlation between $\delta^{65}\text{Cu}$ and Sr-Nd isotopes in the Mesoproterozoic lamproites precludes a $\delta^{65}\text{Cu}$ -low pyroxenite source for these lamproites (Supplementary Fig. S4d). As discussed above, magmatic processes dominated by Ni-bearing sulfide could not cause Cu isotope fractionation, consistent with the BSE-like $\delta^{65}\text{Cu}$ values of mantle pyroxenite xenoliths¹³. Therefore, we

suggest that the metasomatic pyroxenites in the source of the lamproites has the BSE-like source $\delta^{65}\text{Cu}$ value and their homogenous and low $\delta^{65}\text{Cu}$ values are attributed to melting process of a metal-saturated mantle source as presented below and in Fig. 2b. We have revised the manuscript in Lines 154-165.

For the Cu isotope model, which predicts isotopically light melts from a mantle source only when buffered by an Fe-alloy (containing isotopically heavy Cu), I have a couple of questions. The first concerns the assertion that there is no Cu isotope fractionation between sulphides and silicate melt; this is based on the experimental data that show that Ni-rich sulphides and silicates have a negligible Cu isotope fractionation between them. This assertion is used to rule out any Cu isotope fractionation during igneous processes such as sulphide fractionation; although the extant data, which show that MORB and komatiite and also arc lavas all seem to have very predictable Cu isotope compositions seem to suggest that minimal Cu isotope fractionation takes place during sulphide precipitation, I think the authors could have been a bit more careful here when completely ruling this sort of process out.

The first thing to say is that this seems to rely on all sulphides being Ni-rich. This is surely not the case – as the authors know, sulphide petrology and composition can be quite variable, and so simply suggesting that every sulphide involved in the petrogenesis of these rocks is “Ni-rich” is a little difficult to believe based on the evidence provided in this paper; equally, I would ask what “Ni-rich” means – these fractionation factors were taken from a study looking at the crystallisation of the lunar core – are these fractionation factors applicable to Earth’s mantle? This criticism may seem harsh, but all I am saying is that Ni-poor sulphides can fractionate Cu isotopes when equilibrating with a silicate melt – hence I would recommend the authors at least mention this, or take this into account, in their models. I’m pretty sure, when I’ve modelled mantle melting and sulphide precipitation, you don’t get a huge difference in Cu isotope composition in melts, as the maximum sulphide-silicate fractionation factors are pretty small, especially at mantle temperatures.

Reply: We have added Cu isotope fractionation models for magmatic processes using Cu isotope fractionation factors between silicate melt and Ni-free, Ni-bearing sulfides (Lines 678-731). The results indicate that partial melting and magmatic differentiation would not fractionate Cu isotopes of the melts using isotope fractionation factor of Ni-bearing sulfide but would result in heavy Cu isotopes in the melts using isotope fractionation factor of Ni-free sulfide (Fig. 2).

Sulfides in natural peridotite and pyroxenite xenoliths have enriched and highly variable Ni content (5-54 wt.%)^{14,15} (Supplementary Fig. S4). Although the effect of such a large variation of Ni content in sulfide on Cu isotope fractionation factor has not been investigated, the homogeneous $\delta^{65}\text{Cu}$ values of sulfide-saturated MORBs, sulfide-undersaturated komatiites, the bulk silicate Earth^{16,17} (Supplementary Fig. S5), and the lack of Cu isotope fractionation during the evolution of sulfide-saturated arc magmas^{18,19} indicate that most of magmatic processes involving sulfide in the mantle would not cause significant equilibrium Cu isotope fractionation.

Furthermore, the lamproites and nephelinites have higher Ni contents than the MORBs (Supplementary Fig. S4c), indicating that sulfides in participating partial melting and magmatic differentiation are Ni-bearing rather than Ni-free. This is consistent with that their Cu isotope compositions plot away from the modelling results of Cu isotope fractionation during Ni-free sulfide-dominated magmatic processes (Fig. 2b-c).

In relation to the authors' metal-sulphide fractionation models, these are a little difficult to follow, and they aren't described in much detail – the authors mainly refer to their supplementary information database. I think a little more detail on the models in the written component of the supplementary material, plus perhaps

show how [Cu] varies in the melts and restite in the models – as this can help the reader in following the model graphically (as it is, the figure displaying the model is not intuitive because of the buffering of the sulphide in the melt by the metal phase).

Reply: We have added a section “Modelling of Cu isotope fractionation during various magmatic processes” in Methods. We have presented all details of modelling Cu isotope fractionation during partial melting of a peridotite at 1.5 GPa, magmatic differentiation of a MORB melt at 0.2 GPa, and partial melting of a metal-bearing pyroxenite at 6 GPa. See “Methods”, Fig. 2, Supplementary Figs. S6-8, and Supplementary Data S2-4. The Cu contents and isotopes of melts and residues are shown in Supplementary Figs. S6-8.

Finally, could the authors comment on how not all sulphides in mantle rocks are available to melt as well as being less susceptible to metasomatism, i.e. if these are present as inclusions in olivine – is there a potential for a “hidden reservoir” of Cu isotopes in mantle rocks here.

Reply: Most of sulfides in the metasomatic mantle source appear as interstitial crystals between silicate minerals. Furthermore, the mantle source of the lamproites involves an olivine-poor pyroxenite, so that the known sulfide inclusions in olivine (in peridotite) do not apply. Therefore, we will not consider the storage of Cu inclusions in olivine as an important contribution of Cu to the magma.

Minor comments

On line 80, the authors state that the Methods contain sample descriptions and petrology – but really this is limited to the thermobarometry. I think it would be good to include at least a brief description of the rocks utilised in this work, in terms of where and when they were collected, what their mineralogies are, etc.

Reply: We have added a section in “Methods” to describe the geological setting and petrogenesis of these alkaline rocks (Lines 594-646). Furthermore, our recent EPSL paper (<https://doi.org/10.1016/j.epsl.2023.118489>) described these samples in detail.

There seems to be a lot of assumptions/generalisations used in the thermobarometric calculations – whilst I’m sure these are all strong assumptions, it might be worth the authors including some statements to explain how sensitive their P and T estimates are to these assumptions.

Reply: The estimation of the melting temperature and pressure of the alkaline rocks at Aillik Bay was given in detail by Chen et al.⁴ using the method of Sun and Dasgupta²⁰. This thermobarometer treats the Mg-number (Mg#) of olivine in the mantle source and the H₂O content of primary magmas as key parameters²⁰. Chen et al.⁴ compiled the Mg# of olivine of carbonated peridotites and wehrlites to calculate the melting temperature and pressure. However, the Mg# of olivine in the source of phlogopite pyroxenites was not constrained because there is no olivine in the phlogopite pyroxenites (MARID). Chen et al.⁴ assumed 80 for Mg# of olivine in the source of the lamproites, and here, we assume 80, 85, and 90 to evaluate the effect of Mg# of olivine on the calculation of melting temperature and pressure (Supplementary Fig. S2). We found that Mg# of olivine has a small effect on the estimation of melting pressure but has a significant on the estimation of melting temperature. As shown in Supplementary Fig. S2, the melting pressure was about 5-7 GPa for the lamproites and aillikites, and about 2-3 GPa for the nephelinites. The lithosphere thinned over time from >150km (5-7 GPa) for the lamproites at ca. 1.4 Ga and ultramafic lamprophyres/carbonatites at 590-555 Ma to 75-90 km (2-3 GPa) for the nephelinites at 142 Ma.

In figure 4, that authors have a label that states that global plate tectonics starts at 3.2 Ga. Whilst this is one of the periods that is commonly accepted that plate tectonics started globally, perhaps they authors might

just take this off, as this is still a date that is up for debate.

Reply: We have removed it.

I'm not sure that figure 5 in the main text really is adding much to the piece – I think showing an updated Figure S5, which includes [Cu] in the melt and restite, would be more instructive.

Reply: We have added Supplementary Fig. S6-8 to show the modelling details.

Reviewer #3 (Remarks to the Author):

The study of Chen et al. (2023) seeks to constrain the redox evolution of the LAB from the Proterozoic to the present-day using Cu isotopic data from a suite of alkaline rocks located in the NAC of Labrador, Canada. They find a temporal shift in Cu isotopic data that demonstrates that the LAB at this locality has oxidized in the mid-Proterozoic, possibly as a result of the rise of cold, modern style subduction. The premise and topic of this study are both interesting and clever, and for that reason it deserves to be published in this journal, as mantle redox evolution in all its permutations has strong effects on atmospheric evolution. The use of Cu isotopes to quantify mantle redox at the LAB is both innovative and interesting.

However, currently the paper is bogged down by a variety of ideas and exercises that prevent a clear story about Cu isotopes from being imparted to the reader. Chiefly among these is the use of a compilation of a very low number (31 by my counting) of nonorogenic lamproites and a much larger number of carbonatites to postulate a switch from predominant lamproites to predominant carbonatite production. Given the very general title of the paper, the reader is confused and initially thinks that the paper doesn't focus on Cu isotopes of a single locality, and is instead concerned with compiling ages of alkali rocks. The discussion of the age distributions of alkali rocks is treated like a headliner of the paper but it is not, and it should be relegated to the discussion section where the authors try to link the Ailak locality with global change. This brings me to my second overarching problem with the paper: the generalization of a single locality to the whole Earth and the lack of discussion of the NAC specifically in favor of assuming that it reflect global trends. The mantle lithosphere shows immense amounts of redox heterogeneity relating to metasomatism, and a global secular trend is not demonstrated by a trend in a single 20x20 km area. If the authors were to alter the title and discussion to instead discuss how their data may be indicative of a global change as opposed to definitively stating that these data show global change, the manuscript would be much stronger. The method used here (Cu isotopes of alkali rocks) can be concluded to show promise for quantifying global trends once more localities are analyzed, and that in my opinion is the most important conclusions of the paper.

Reply: We have weakened the reliance on evidence from the distribution of lamproites. We have compared our results with the redox state of cratonic peridotite xenoliths (Lines 209-228). See responses to Reviewer 1 concerning the use of peridotite data from other cratons.

Two more general points I have with the study: how easy is it to distinguish orogenic from nonorogenic lamproites in tectonically complex Proterozoic terranes? Is it possible that misidentification could have lead to the distribution in nonorogenic lamproites observed? Finally, much is made of the globally rare nature of carbonatites in the Precambrian, but the low preservation potential of carbonatites in the rock record is never once mentioned. This is an oversight and should be corrected, as given the low amount of crust available from the early Earth and the rarity of carbonatites in general throughout Earth history, the exponential

increase curve of carbonatites shown must be placed in proper context.

Reply: We compile the global age distribution of nonorogenic lamproites (Supplementary Data S1 and Fig. 1c-d). The compiled nonorogenic lamproites occur in stable craton regions (Supplementary Data S1), and are much younger than formation of cratons, indicating that they have no relationship with subduction and collision. These nonorogenic lamproites originated from the lower lithosphere mantle and were formed by melting in reduced conditions²¹. In contrast, orogenic lamproites occur in mobile belts being generated at shallow depths in post-collisional environments, and so are not relevant for the lower cratonic mantle²². The result shows that the nonorogenic lamproites mainly occurred in the early-middle Proterozoic and Mesozoic-Cenozoic period, with a conspicuous gap between 1000 and 300 Ma (Fig. 1b).

The age distribution of carbonatites and kimberlites are from Woolley and Kjarsgaard²³ and Tappe et al.²⁴, respectively. The abundance of the carbonatites and kimberlites show a general increase since ~2 Ga with a significant increase over the past 700 Ma. We have added text based on these two review papers which shows that preservation bias is not a major problem. We have revised the manuscript in Lines 657-677.

Overall, the paper is well written and the figures are thorough and highly professional. Re-focusing the paper on Cu isotopic data for NAC lavas and backing away from global generalizations and the age compilation data will serve this excellent dataset better.

Reply: Thanks for your constructive suggestions.

Thanks for an interesting read,

Specific Comments

-In this section I detail specific in-line comments. I have also provided numerous grammatical and usage edits in the “tracked changes” file returned with this review.

-In Line 135, V/Sc is used as a proxy for oxygen fugacity of low melt fraction, enriched lavas. V/Sc is only a useful oxybarometer if three conditions are met: 1) melting does not leave garnet in the residue or fractionate garnet, due to the strongly compatible nature of Sc in garnet (much like the HREE) relative to V, 2) the degree of partial melting is well constrained and similar between the two types of rocks being compared. At low degrees of partial melting (<10%) like those examined here, small changes in the degree of partial melting can strongly influence V/Sc at constant fO_2 (see Figure 1 of Li and Lee, 2004), 3) Finally and most importantly, V/Sc is a forward model for fO_2 and assumes a constant V/Sc in the mantle source regions of the rocks examined. This is almost certainly untrue in this case as the mantle source regions are highly metasomatized lithosphere. I would not use V/Sc as a redox proxy at all, as your Cu isotope redox story is compelling enough without it.

Reply: V/Sc ratio is not used to reflect the source but represent the fO_2 in magma evolution. The evolved damtjernites and carbonatites are thought to have formed by liquid immiscibility from aillikite magma at a shallow crustal depth⁶. The evolved damtjernites and carbonatites show higher and highly variable V/Sc ratios than primary aillikite magmas (Supplementary Fig. S3) which may be the result of variable fO_2 during magmatic differentiation. The positive correlation between V/Sc and $\delta^{65}Cu$ values in the evolved

damtjernites and carbonatites (Supplementary Fig. S3) reveals oxidization of some Cu^{1+} to Cu^{2+} in the magmas during liquid immiscibility, which is supported by the occurrence of sulfate crystals in carbonatites⁶.

-In Line 145 you use Cu/Sc to argue that mantle source regions do not show prior Cu enrichments. You use the observation that Cu/Sc is elevated in OIB relative to MORB to invoke Cu enrichment in the OIB source mantle, but similar to my previous comment, this is almost certainly the result of the compatibility of Sc in garnet relative to Cu and unrelated to Cu enrichment, as garnet is found in the residue of most OIB but not MORB. If you want a trivalent cation that is less influenced by garnet, try using Ga, as V/Ga has been suggested for use instead of V/Sc for this very reason (Laubier XXX). Alternatively, all of the V/Sc and Cu/Sc discussion can be cut and the manuscript focused better on the main point: Cu isotopes.

Reply: We have deleted the discussion of Cu/Sc.

-Line 159: How did you chose your starting isotopic value? Is it the accepted BSE value? If so cite it.

Reply: We have cited it.

-Line 178: What is NAC? Define terms appropriately.

Reply: We have defined NAC in Line 57.

-Line 182: What about the youngest rocks and their $f\text{O}_2$? Tell the three stages of the story here, not just the first two.

Reply: The $f\text{O}_2$ calculation using the method of Stagno et al.²⁵ requires primary melt compositions, especially CO_2 content. The 142 Ma nephelinites experienced significant fractional crystallization⁴. Furthermore, they are CO_2 -bearing magmas and may experience degassing during eruption. Therefore, it is difficult to calculate the $f\text{O}_2$ of the nephelinites.

Line 191: What about preservation bias of carbonatite rocks? This is never mentioned or addressed. Carbonatite rocks can weather much more readily than silicates and therefore their preservation from the Archean to present day is perhaps unlikely.

Reply: We have now discussed this point in “Methods” and show, based on the compilations of Woolley and Kjarsgaard²³ and Tappe et al.²⁴, that preservation bias is not a major problem.

-Line 222: This almost certainly testably by examining initial Nd and Sr isotopic signatures of global carbonatites and lamproites, as subducted recycled materials should impart a strongly enriched signature. I am not suggesting that you go through the exercise of compiling these data and seeing if such a shift is evident, but I am saying you should state that this would be a good way to test your model in a future study. Besides it is always good practice to end a paper in an open-ended manner, with an eye towards what sort of future data or studies could support the model presented.

Reply: We have discussed this point in our revised manuscript. We agree that it is difficult to decode the oxidation mechanism from our data.

-Figure 4: Take out “vast” from the carbonatite-in line

Reply: We have deleted it.

-Figure 4: Why is the “onset of plate tectonics” marked on this figure? This is never discussed in the text and

remains a hotly debated topic. This figure is busy, and this label adds little to it. As you only focus on the Proterozoic to Phanerozoic, why not start the figure at 2500 Ma?

Reply: We have deleted it.

-Figure 5: this is an excellent figure for illustrating your model for LAB, good job.

Reply: Thanks a lot.

-Line 532: Which is it? 1000 or 2000 ppm or did you vary the concentration between these two values?

Reply: We have rephrased it.

References:

- 1 O'Hara, M. J. The bearing of phase equilibria studies in synthetic and natural systems on the origin and evolution of basic and ultrabasic rocks. *Earth-Science Reviews* **4**, 69-133 (1968).
- 2 Sarkar, S. *et al.* Sublithospheric melt input in cratonic lamproites. *Geology* **50**, 1296-1300 (2022).
- 3 Sarkar, S. *et al.* Derivation of Lamproites and Kimberlites from a Common Evolving Source in the Convective Mantle: the Case for Southern African 'Transitional Kimberlites'. *Journal of Petrology* **64** (2023).
- 4 Chen, C. *et al.* Calcium isotopes track volatile components in mantle sources of alkaline rocks and associated carbonatites. *Earth and Planetary Science Letters* **625**, 118489 (2024).
- 5 Tappe, S. *et al.* Craton reactivation on the Labrador Sea margins: $^{40}\text{Ar}/^{39}\text{Ar}$ age and Sr–Nd–Hf–Pb isotope constraints from alkaline and carbonatite intrusives. *Earth and Planetary Science Letters* **256**, 433-454 (2007).
- 6 Tappe, S. *et al.* Genesis of Ultramafic Lamprophyres and Carbonatites at Aillik Bay, Labrador: a Consequence of Incipient Lithospheric Thinning beneath the North Atlantic Craton. *Journal of Petrology* **47**, 1261-1315 (2006).
- 7 Yaxley, G. M., Berry, A. J., Rosenthal, A., Woodland, A. B. & Paterson, D. Redox preconditioning deep cratonic lithosphere for kimberlite genesis – evidence from the central Slave Craton. *Scientific Reports* **7**, 30 (2017).
- 8 Tappe, S. *et al.* Sheared Peridotite and Megacryst Formation Beneath the Kaapvaal Craton: a Snapshot of Tectonomagmatic Processes across the Lithosphere–Asthenosphere Transition. *Journal of Petrology* **62**, 1-39 (2021).
- 9 Foley, S. F., Ezad, I. S., van der Laan, S. R. & Pertermann, M. Melting of hydrous pyroxenites with alkali amphiboles in the continental mantle: 1. Melting relations and major element compositions of melts. *Geoscience Frontiers* **13**, 101380 (2022).
- 10 Maréchal, C. N., Télouk, P. & Albarède, F. Precise analysis of copper and zinc isotopic compositions by plasma-source mass spectrometry. *Chemical Geology* **156**, 251-273 (1999).
- 11 Zhu, Y. *et al.* High-precision Copper and Zinc Isotopic Measurements in Igneous Rock Standards Using Large-geometry MC-ICP-MS. *Atomic Spectroscopy* **40**, 6 (2019).
- 12 Liu, S.-A. *et al.* High-precision copper and iron isotope analysis of igneous rock standards by MC-ICP-MS. *Journal of Analytical Atomic Spectrometry* **29**, 122-133 (2014).
- 13 Zhang, G. *et al.* Copper mobilization in the lower continental crust beneath cratonic margins, a Cu isotope perspective. *Geochimica et Cosmochimica Acta* **322**, 43-57 (2022).
- 14 Aulbach, S. *et al.* Mantle formation and evolution, Slave Craton: constraints from HSE abundances and Re–

- Os isotope systematics of sulfide inclusions in mantle xenocrysts. *Chemical Geology* **208**, 61-88 (2004).
- 15 Aulbach, S. *et al.* Sulfide and whole rock Re–Os systematics of eclogite and pyroxenite xenoliths from the Slave Craton, Canada. *Earth and Planetary Science Letters* **283**, 48-58 (2009).
 - 16 Savage, P. S. *et al.* Copper isotope evidence for large-scale sulphide fractionation during Earth's differentiation. *Geochemical Perspectives Letters* **1**, 53-64 (2015).
 - 17 Liu, S.-A. *et al.* Copper isotopic composition of the silicate Earth. *Earth and Planetary Science Letters* **427**, 95-103 (2015).
 - 18 Wang, Z. *et al.* Copper recycling and redox evolution through progressive stages of oceanic subduction: Insights from the Izu-Bonin-Mariana forearc. *Earth and Planetary Science Letters* **574**, 117178 (2021).
 - 19 Wang, Z. *et al.* Evolution of copper isotopes in arc systems: Insights from lavas and molten sulfur in Niutahi volcano, Tonga rear arc. *Geochimica et Cosmochimica Acta* **250**, 18-33 (2019).
 - 20 Sun, C. & Dasgupta, R. Thermobarometry of CO₂-rich, silica-undersaturated melts constrains cratonic lithosphere thinning through time in areas of kimberlitic magmatism. *Earth and Planetary Science Letters* **550**, 116549 (2020).
 - 21 Foley, S. F. Experimental constraints on phlogopite chemistry in lamproites: 1. The effect of water activity and oxygen fugacity. *European Journal of Mineralogy* **1**, 411-426 (1989).
 - 22 Prelević, D., Foley, S., Romer, R. & Conticelli, S. Mediterranean Tertiary lamproites derived from multiple source components in postcollisional geodynamics. *Geochimica et Cosmochimica Acta* **72**, 2125-2156 (2008).
 - 23 Woolley, A. & Kjarsgaard, B. *Carbonatite occurrences of the world: map and database*, (Geological Survey of Canada, 2008) http://geopub.nrcan.gc.ca/moreinfo_e.php?id=225115&_h=Woolley.
 - 24 Tappe, S., Smart, K., Torsvik, T., Massuyeau, M. & de Wit, M. Geodynamics of kimberlites on a cooling Earth: Clues to plate tectonic evolution and deep volatile cycles. *Earth and Planetary Science Letters* **484**, 1-14 (2018).
 - 25 Stagno, V. & Frost, D. J. Carbon speciation in the asthenosphere: Experimental measurements of the redox conditions at which carbonate-bearing melts coexist with graphite or diamond in peridotite assemblages. *Earth and Planetary Science Letters* **300**, 72-84 (2010).

REVIEWER COMMENTS

Reviewer #1 (Remarks to the Author):

I have read the authors' replies to all reviewer comments and find them to be reasonably satisfactory, although I am not entirely convinced of all the arguments presented, in particular regarding the nature of sulphides, in particular of metasomatic origin, in the cratonic lithospheric mantle. However, this can be fixed by putting an appropriate caveat and replacing a few "indicates" with "may indicate", and is not a reason to hold up publication of this manuscript.

The authors made significant changes in response to the reviewer comments and provide welcome clarifications in the supplement. They may wish to consider the following very minor issues, which came up when I read the version without tracked changes (numbers refer to lines in this clean document) with a "fresh eye":

12 the abstract really should mention the actual Cu isotope values prior and post oxidation to do justice to the title and the novelty of the approach to estimating fO_2

57 could simply say "is a representative" to avoid the repetition

80 use the umlaut "wüstite"

98 perhaps more accurate to say "no resolvable Cu isotopic fractionation"

122 "suggesting" might be a better word. Lorand and Gregoire 2002 CMP show that sulphide are not in equilibrium with peridotites with respect to Ni/Fe, having lower Ni/Fe

143 "oxidation", not "oxidisation" - multiple instances in manuscript

153 fix the the

160 "low- $\delta^{65}\text{Cu}$ "

186 "with neither.... nor"

232 delete "to"

258 perhaps refer to "isolated" or "regional" oxidation or similar, to provide a contrast with "large-scale" mentioned in the second part of the sentence?

269 I suggest to replace "An alternative interpretation" with "One interpretation". The reason is that (1) you don't mention the interpretation of refs 51, 52, which does in fact differ from the one you propose, and (2) 51, 52 proposed a scenario that can explain the observed secular increase in atmospheric oxygenation. Your interpretation cannot because the oxidised components that you recycle would have

been taken from the ocean-atmosphere system, causing a DECREASE in atmospheric redox budget, thereby delaying its oxygenation if not compensated by other processes. There is absolutely no need to go into these details, but I wish to alert you to the ulterior implications of the scenarios that are proposed to explain the proposed increase in asthenospheric fO_2 , and why I suggest the change in wording.

344 Dr., not Prof.

I think this is now in a fine state, and with very moderate additional effort will be ready for publication.

Reviewer #2 (Remarks to the Author):

Review of “Copper isotopes track the Neoproterozoic oxidation of cratonic mantle roots”

I thank the authors for their detailed rebuttal document to the previous round of reviews, as well as providing a tracked-changes document. Broadly, this is a much improved manuscript that I would be happy to recommend for publication, perhaps after a couple of minor alterations/clarifications as noted below.

I think changing the focus of the paper to suggesting Cu isotopes as a proxy for mantle melting fO_2 , and then moving on to a wider discussion of the oxidation of the NAC from other proxies makes the work much more holistic and less reliant on the smallish database of Cu isotope data, which is excellent. I do agree with another of the reviewers that the authors could lean even more into the novelty of using Cu isotopes here to show that their lamproites formed with a metal alloy in the source region. This is (I think) the first study to suggest or even attempt to use copper isotopes in this way, and I think more should be made of this – particularly given what seems to be a set of data that really do agree with the authors hypothesis.

For instance, Line 73: In this paragraph, I would highlight that although Cu isotopes are redox sensitive they have not really been applied to investigating the redox conditions of mantle melting – this is the authors’ new and novel idea. At the moment, in the paragraph, it sounds like Cu isotopes are a routine tracer – but they are not really (there have been attempts but not, I think, as successful as this work). This should be the hook that gets people interested in the paper. Furthermore, at the end of this section, in Line 85: perhaps a summary sentence of the proxy here – i.e. “Hence, if a melt is generated with metal alloy in the source, the copper isotope composition of that melt would be lighter than BSE”. This novelty could be further highlighted in the abstract, and also in a summary sentence at the end of the main text, e.g.: perhaps again point to the potential for Cu isotopes to apparently be diagnostic of melting in the presence of metal alloys – a robust proxy for reduced melts – could be interesting not just for mantle petrology but for studies of reduced bodies in the solar system.

I have one or two questions that have dawned on my based on my second reading of this work, which I

think will be easy to address:

The first (Line 176): is there any other chemical evidence in the Mesoproterozoic lamproites that there was a metal alloy in the melt source? How about other siderophile elements which might show depletions given they would be strongly enriched in this alloy relative to the silicate minerals that are fusing to form the melt? Again, this would be another line of evidence to support the interpretation of the Cu isotope data.

The second again refers to the argument that metasomatic alteration of the source is not the reason for the isotopically light Cu of the lamproites; (Line 160) Could the authors explain in a bit more detail as to why this lack of correlation between Nd isotopes and Cu isotopes rules out the idea that a metasomatism could have affected the Cu isotope composition of the source rock? I don't really follow this logic (I'm probably being stupid!).

Does ruling out a wholly pyroxenitic source strictly rule out the potential for cryptic metasomatism in the source which may (or may not) have altered its Cu isotope composition? The only reason I ask is that biotite mica can display mineral-melt D values >1 in some systems; I'm not sure there's much in the way of phlogopite Cu partitioning data unfortunately, but if the lamproites are forming from the breakdown of phlogopite, could this mica have enriched Cu and then released its cargo (which could be isotopically light, who knows?) to the melt?

Anyway, I thank the authors again for their conscientious approach to the first round of reviews. My main suggestions would be to make the novelty of the Cu isotope approach more overt in the work, and then perhaps spell out more clearly the arguments against metasomatic alteration of the source as the cause of the isotopically light Cu in the lamproites (and potentially highlight any other chemical evidence for metal alloy in the source). Other than that I would be happy to recommend publication.

Some minor points:

Line 57: repetition: "...represents a representative..."

Line 90: Perhaps a reference to the MORB reference values here?

Line 105: Here perhaps qualify this new statement to limit it to (mantle) partial melting processes – there's fairly good evidence that crystallisation of sulphides in e.g. magma chambers or subarc mantle can affect the Cu isotope compositions of melts – even if basalts (*sensu lato*) don't seem to reflect this globally. N.B. any effect of fractional crystallisation of sulphides in MORB melt is likely buffered by continuous refreshing of the magma system by ongoing melting.

Line 131: I'm not sure Figure 3 best illustrates Cu behaving as an incompatible element in komatiites, ultramafic lamprophyres etc. Nevertheless, this section could be clarified with a simple statement that states that without sulphides on the liquidus, Cu will behave as a strongly incompatible element and there should be no a priori reason to suggest that magmatic differentiation will affect the Cu isotope composition of a sulphide-undersaturated melt.

Line 147: Perhaps also point the readers to the compilation of arc basalts some of which are pushed to heavier Cu isotope compositions, so show this effect of dehydration and release of isotopically heavy Cu.

Line 222: sp. "peridotite"

Line 607: Perhaps point to references here where readers may find more information about the rocks in question.

Figure 2 – perhaps include the refs for the metal-silicate fractionation factors too, if you are doing it for

the sulphide-silicate factors.

Fig 6. Spelling mistake in lamproites (in the key). Also spelling mistake in the caption (curce instead of "curve").

Reviewer #3 (Remarks to the Author):

Please see attached review.

Thank you for your extensive re-writes, and especially for removing much of the global focus of the paper to focus instead specifically on the rocks you analyzed. I think the paper is much stronger as a result of the edits. I have gone through again and have a few additional minor comments, but overall I think the paper is ready to go once these issues are fixed.

Specific Comments

Line 57: **is** a representative

Line 114: what do you mean by “residual” melts? Do you mean residues?

Line 140: add a line here saying why you know higher V/Sc in carbonatites means higher fO_2 : i.e. oxidized V^{+4} has a greater affinity for carbonatite melts than Sc^{+3} .

Line 149: **our modeling shows that** addition of crustal materials...

Line 197: that **the** fO_2 of...

Line 208: Please proof-read this section once more, there are several grammatical errors.

Line 221: Use “log units” instead of “orders of magnitude” for oxygen fugacity

Line 222: spelling typo

Line 232: delete “to”

Line 246: **oxidation state** has continually increased since...

Line 261: as **measomatic** agents...

Line 262: delete “the”

Line 270: just use “colder” as opposed to “cold and warm”

Response to referees' comments, manuscript NCOMMS-23-32963A

Chunfei Chen, Stephen F. Foley, Svyatoslav S. Shcheka, Yongsheng Liu

"Copper isotopes track the Neoproterozoic oxidation of cratonic mantle roots"

Referees' comments in black

Replies in blue

We thank the referees for the constructive comments and suggestions that significantly improved the manuscript.

REVIEWER COMMENTS

Reviewer #1 (Remarks to the Author):

I have read the authors' replies to all reviewer comments and find them to be reasonably satisfactory, although I am not entirely convinced of all the arguments presented, in particular regarding the nature of sulphides, in particular of metasomatic origin, in the cratonic lithospheric mantle. However, this can be fixed by putting an appropriate caveat and replacing a few "indicates" with "may indicate", and is not a reason to hold up publication of this manuscript.

The authors made significant changes in response to the reviewer comments and provide welcome clarifications in the supplement. They may wish to consider the following very minor issues, which came up when I read the version without tracked changes (numbers refer to lines in this clean document) with a "fresh eye":

12 the abstract really should mention the actual Cu isotope values prior and post oxidation to do justice to the title and the novelty of the approach to estimating fO_2

Reply: We have revised the abstract with actual Cu isotope values of the lamproites and <0.59 Ga silica-undersaturated alkaline rocks (Lines 18-20).

57 could simply say "is a representative" to avoid the repetition

Reply: We have revised it (Line 56).

80 use the umlaut "wüstite"

Reply: Done (Line 79).

98 perhaps more accurate to say "no resolvable Cu isotopic fractionation"

Reply: We have revised it (Line 99).

122 "suggesting" might be a better word. Lorand and Gregoire 2002 CMP show that sulphide are not in equilibrium with peridotites with respect to Ni/Fe, having lower Ni/Fe

Reply: We have revised it (Line 122).

143 "oxidation", not "oxidisation" - multiple instances in manuscript

Reply: We have revised it all through the manuscript (Line 145).

153 fix the the

Reply: We have deleted the repetitive “the”.

160 "low-d65Cu"

Reply: We have revised it (Line 163).

186 "with neither.... nor"

Reply: We have revised it (Line 190).

232 delete "to"

Reply: Done.

258 perhaps refer to “isolated” or “regional” oxidation or similar, to provide a contrast with “large-scale” mentioned in the second part of the sentence?

Reply: We have revised it (Line 261).

269 I suggest to replace “An alternative interpretation” with “One interpretation”. The reason is that (1) you don’t mention the interpretation of refs 51, 52, which does in fact differ from the one you propose, and (2) 51, 52 proposed a scenario that can explain the observed secular increase in atmospheric oxygenation. Your interpretation cannot because the oxidised components that you recycle would have been taken from the ocean-atmosphere system, causing a DECREASE in atmospheric redox budget, thereby delaying its oxygenation if not compensated by other processes. There is absolutely no need to go into these details, but I wish to alert you to the ulterior implications of the scenarios that are proposed to explain the proposed increase in asthenospheric fO₂, and why I suggest the change in wording.

Reply: We have revised it (Line 272).

344 Dr., not Prof.

Reply: We have revised it (Line 350).

I think this is now in a fine state, and with very moderate additional effort will be ready for publication.

Thanks for your constructive suggestions.

Reviewer #2 (Remarks to the Author):

Review of “Copper isotopes track the Neoproterozoic oxidation of cratonic mantle roots”

I thank the authors for their detailed rebuttal document to the previous round of reviews, as well as providing a tracked-changes document. Broadly, this is a much improved manuscript that I would be happy to recommend for publication, perhaps after a couple of minor alterations/clarifications as noted below.

I think changing the focus of the paper to suggesting Cu isotopes as a proxy for mantle melting fO₂, and then moving on to a wider discussion of the oxidation of the NAC from other proxies makes the work much more holistic and less reliant on the smallish database of Cu isotope data, which is excellent. I do agree with another of the reviewers that the authors could lean even more into the novelty of using Cu isotopes here to show that their lamproites formed with a metal alloy in the source region. This is (I think) the first study to suggest or even attempt to use copper isotopes in this way, and I think more should be made of this – particularly given what seems to be a set of data that really do agree with the authors hypothesis.

For instance, Line 73: In this paragraph, I would highlight that although Cu isotopes are redox sensitive they have not really been applied to investigating the redox conditions of mantle melting – this is the authors' new and novel idea. At the moment, in the paragraph, it sounds like Cu isotopes are a routine tracer – but they are not really (there have been attempts but not, I think, as successful as this work). This should be the hook that gets people interested in the paper. Furthermore, at the end of this section, in Line 85: perhaps a summary sentence of the proxy here – i.e. “Hence, if a melt is generated with metal alloy in the source, the copper isotope composition of that melt would be lighter than BSE”. This novelty could be further highlighted in the abstract, and also in a summary sentence at the end of the main text, e.g.: perhaps again point to the potential for Cu isotopes to apparently be diagnostic of melting in the presence of metal alloys – a robust proxy for reduced melts – could be interesting not just for mantle petrology but for studies of reduced bodies in the solar system.

Reply: Thanks for your suggestion. We have reinforced the statement on the novelty of using copper isotopes to trace redox processes in the mantle (Lines 74 and 83-85), and highlighted this further at the end of the main text (Lines 277-280).

I have one or two questions that have dawned on my based on my second reading of this work, which I think will be easy to address:

The first (Line 176): is there any other chemical evidence in the Mesoproterozoic lamproites that there was a metal alloy in the melt source? How about other siderophile elements which might show depletions given they would be strongly enriched in this alloy relative to the silicate minerals that are fusing to form the melt? Again, this would be another line of evidence to support the interpretation of the Cu isotope data.

Reply: Unfortunately, the PGE contents of these samples have not been reported. But it would be good to embark on a new project about the PGE contents of these samples.

The second again refers to the argument that metasomatic alteration of the source is not the reason for the isotopically light Cu of the lamproites; (Line 160) Could the authors explain in a bit more detail as to why this lack of correlation between Nd isotopes and Cu isotopes rules out the idea that a metasomatism could have affected the Cu isotope composition of the source rock? I don't really follow this logic (I'm probably being stupid!).

Reply: We have clarified how the lack of correlations between $\delta^{65}\text{Cu}$ and Sr-Nd isotopes in the Mesoproterozoic lamproites preclude a low- $\delta^{65}\text{Cu}$ pyroxenite source for these lamproites (Lines 159-162). Melting of a low- $\delta^{65}\text{Cu}$ pyroxenite vein would cause melting of and reaction with the surrounding peridotite wall, which could produce a melt with a mixed isotope signature blending characteristics between mantle peridotite with BSE-like $\delta^{65}\text{Cu}$ and high $^{143}\text{Nd}/^{144}\text{Nd}$, and metasomatic pyroxenite with low $\delta^{65}\text{Cu}$ and low $^{143}\text{Nd}/^{144}\text{Nd}$.

Does ruling out a wholly pyroxenitic source strictly rule out the potential for cryptic metasomatism in the

source which may (or may not) have altered its Cu isotope composition? The only reason I ask is that biotite mica can display mineral-melt D values >1 in some systems; I'm not sure there's much in the way of phlogopite Cu partitioning data unfortunately, but if the lamproites are forming from the breakdown of phlogopite, could this mica have enriched Cu and then released its cargo (which could be isotopically light, who knows?) to the melt?

Reply: Regrettably, our search found no data on Cu partitioning between phlogopite and silicate melts in the mantle. Nonetheless, given its silicate mineral structure, phlogopite is unlikely to exhibit high levels of Cu enrichment. Historically, silicate minerals have not been regarded as Cu-rich in the majority of existing literature and unpublished data in our research group show that phlogopite may be enriched in Ni and Cr, but not Cu. Additionally, the lamproites we examined display characteristics of sulfide saturation along with low Cu content, contradicting the notion of melting Cu-rich phlogopite.

Anyway, I thank the authors again for their conscientious approach to the first round of reviews. My main suggestions would be to make the novelty of the Cu isotope approach more overt in the work, and then perhaps spell out more clearly the arguments against metasomatic alteration of the source as the cause of the isotopically light Cu in the lamproites (and potentially highlight any other chemical evidence for metal alloy in the source). Other than that I would be happy to recommend publication.

Reply: Thanks.

Some minor points:

Line 57: repetition: "...represents a representative..."

Reply: We have revised it (Line 56).

Line 90: Perhaps a reference to the MORB reference values here?

Reply: The reference of Savage et al.¹ is already there.

Line 105: Here perhaps qualify this new statement to limit it to (mantle) partial melting processes – there's fairly good evidence that crystallisation of sulphides in e.g. magma chambers or subarc mantle can affect the Cu isotope compositions of melts – even if basalts (sensu lato) don't seem to reflect this globally. N.B. any effect of fractional crystallisation of sulphides in MORB melt is likely buffered by continuous refreshing of the magma system by ongoing melting.

Reply: Cu isotopic fractionation in mantle-derived rocks has been observed in certain cumulate rocks, like gabbroic rocks^{2,3}, but not in their corresponding magmas. Yet, these Cu isotopic variations are not believed to arise from magmatic processes occurring during the cumulus stage³. Rather, the significant Cu isotopic variation in gabbroic rocks is ascribed to post-cumulus melt evolution in the lower crust, which involves fractionation between sulfide and inter-cumulus melts at relatively low temperatures at the hand specimen scale³.

Line 131: I'm not sure Figure 3 best illustrates Cu behaving as an incompatible element in komatiites, ultramafic lamprophyres etc. Nevertheless, this section could be clarified with a simple statement that states that without sulphides on the liquidus, Cu will behave as a strongly incompatible element and there should be no a priori reason to suggest that magmatic differentiation will affect the Cu isotope composition of a sulphide-undersaturated melt.

Reply: We have added a statement for this point (Lines 131-133).

Line 147: Perhaps also point the readers to the compilation of arc basalts some of which are pushed to heavier Cu isotope compositions, so show this effect of dehydration and release of isotopically heavy Cu.

Reply: We propose that the magmatic differentiation of damtjernites and carbonatites in the rifting setting diverges from that of arc basalts. We therefore refrain from discussing the processes and Cu isotopes specific to arc basalts.

Line 222: sp. “peridotite”

Reply: We have revised it.

Line 607: Perhaps point to references here where readers may find more information about the rocks in question.

Reply: We have added the references (Line 615).

Figure 2 – perhaps include the refs for the metal-silicate fractionation factors too, if you are doing it for the sulphide-silicate factors.

Reply: We have added the references in Figure 2.

Fig 6. Spelling mistake in lamproites (in the key). Also spelling mistake in the caption (curce instead of “curve”).

Reply: We have revised these typos.

Reviewer #3 (Remarks to the Author):

Thank you for your extensive re-writes, and especially for removing much of the global focus of the paper to focus instead specifically on the rocks you analyzed. I think the paper is much stronger as a result of the edits. I have gone through again and have a few additional minor comments, but overall I think the paper is ready to go once these issues are fixed.

-Robert Nicklas

Boston College

Thank you very much for the constructive comments.

Specific Comments

Line 57: is a representative

Reply: We have revised it (Line 56).

Line 114: what do you mean by “residual” melts? Do you mean residues?

Reply: We have revised it (Line 115).

Line 140: add a line here saying why you know higher V/Sc in carbonatites means higher fO_2 : i.e. oxidized V+4 has a greater affinity for carbonatite melts than Sc+3.

Reply: We have added it (Line 143).

Line 149: our modeling shows that addition of crustal materials...

Reply: We have clarified it (Line 151).

Line 197: that the fO_2 of...

Reply: We have revised it (Line 201).

Line 208: Please proof-read this section once more, there are several grammatical errors.

Reply: We have revised it.

Line 221: Use “log units” instead of “orders of magnitude” for oxygen fugacity

Reply: We have revised it (Line 225).

Line 222: spelling typo

Reply: We have revised it.

Line 232: delete “to”

Reply: We have deleted it.

Line 246: oxidation state has continually increased since...

Reply: We have revised it (Line 250-251).

Line 261: as measomatic agents...

Reply: We have revised it (Line 265).

Line 262: delete “the”

Reply: We have deleted it.

Line 270: just use “colder” as opposed to “cold and warm”

Reply: We have revised it (Line 274).

References:

- 1 Savage, P. S. *et al.* Copper isotope evidence for large-scale sulphide fractionation during Earth’s differentiation. *Geochemical Perspectives Letters* **1**, 53-64 (2015).
- 2 Zou, Z. *et al.* Contrasting Cu isotopes in mid-ocean ridge basalts and lower oceanic crust: Insights into the oceanic crustal magma plumbing systems. *Earth and Planetary Science Letters* **627**, 118563 (2024).
- 3 Zhang, W.-Q., Liu, C.-Z., Johan Lissenberg, C. & Li, X.-N. Post-cumulus control on copper isotopic fractionation during oceanic intra-crustal magmatic differentiation. *Geochimica et Cosmochimica Acta* **369**, 35-50 (2024).

REVIEWERS' COMMENTS

Reviewer #2 (Remarks to the Author):

Dear Editor,

Thanks again for providing me with the opportunity to review this work. The responses to my additional comments from my second review have been broadly addressed in the revised version. The only minor thing which I would recommend, which the authors did not address, would be to improve the wording of the abstract to better “advertise” the findings of the study, and the utilisation of the Cu isotope proxy – such as:

Line 15: I would rewrite this to be: “In this study we employ a novel proxy – Cu isotopes, which are redox-sensitive – to Mesoproterozoic lamproites and younger silica-undersaturated alkaline rocks sourced from the LAB of the North Atlantic Craton...”. This really highlights the novelty of the study. Furthermore, I would add an extra sentence before the one starting Line 15, explaining why it is important to understand the redox evolution of the LAB.

Otherwise, I have no further comments to make and would be happy for this work to be published in Nature Comms.

Line 20: Rewrite: “...indicate that by the end of the Proterozoic, the LAB under the North Atlantic Craton had become more oxidised, stabilising CO₂ and H₂O and destabilising metals”.

Line 24: Final line in the abstract “...this oxidation might occur in global cratonic roots” – I think I know what the authors are trying to say here, but it might be better worded as “our study adds to the evidence that oxidation of craton roots was a global event, which may have occurred at approximately the same time”.

Line 74: “employing novel copper isotopes” is not a good way to word this statement – the copper isotopes aren’t novel, it’s the use of them in this manner that is – e.g. “by employing a novel proxy: that of copper isotopes...”

Line 115: “heavy Cu isotope compositions in the melts”

Line 277: Perhaps delete “clear” here – as there are other potential ways to fractionate Cu isotopes (which need to be ruled out, as this study does) before they can be assigned to reflecting metallic Cu in the source.